# In-Context Generation with Regional Constraints
# for Instructional Video Editing

**Zhongwei Zhang** [1] [†]  **Fuchen Long** [2]  **Wei Li** [1]  **Zhaofan Qiu** [2]  **Wu Liu** [1]  **Ting Yao** [2]  **Tao Mei** [2]

## Abstract

The in-context generation paradigm has demonstrated strong power in instructional image editing for better synthesis quality. Nevertheless, shaping such in-context learning for instructional video editing is not trivial. Without specifying editing regions, the results can suffer from the issue of inaccurate editing regions and the token interference between different areas. To address these, we present ReCo, a new instructional video editing paradigm that novelly delves into **Re**gional **Co**nstraint modeling between editing and non-editing areas. Technically, ReCo width-wise concatenates source and target video for joint denoising. In model training, ReCo formulates regional constraints with two regularization terms, i.e., latent and attention regularization, on one-step backward denoised latents and attention maps, respectively. The former increases the latent discrepancy of the editing region between source and target videos while reducing that of non-editing areas, emphasizing editing area modification and alleviating unexpected content generation. The latter suppresses the attention of tokens in the editing region to the tokens in counterpart of the source video, thereby mitigating their interference during novel object generation in target video. Furthermore, we propose a large-scale, high-quality video editing dataset, i.e., ReCo-Data, comprising 500K instruction-video pairs. Extensive experiments conducted on four major instruction-based video editing tasks verify the superiority of ReCo. Code is available at https://github.com/HiDream-ai/ReCo.

[†] Work done during an internship at HiDream.ai Inc. [1]University of Science and Technology of China, Anhui, China [2]HiDream.ai Inc, Beijing, China. Correspondence to: Zhaofan Qiu <qiuzhaofan@hidream.ai>, Wu Liu <liuwu@ustc.edu.cn>.

*Proceedings of the 43rd International Conference on Machine Learning*, Seoul, South Korea. PMLR 306, 2026. Copyright 2026 by the author(s).

## 1. Introduction

With the rapid advancements in generative models (Ho et al., 2020; Nichol et al., 2022; Esser et al., 2024; Yang et al., 2024; Zhang et al., 2025d; Li et al., 2025a; Liang et al., 2025; Zhang et al., 2025a; Chen et al., 2025b;a; Li et al., 2026; Long et al., 2024; Liu et al., 2025a; 2024a; 2026; Li et al., 2025c), instruction-based visual editing for both image and video has garnered attention. Recent instruction-based image editing models (Zhang et al., 2025f; Labs et al., 2025) are capable of editing input images based on natural language instructions without additional condition. Nevertheless, replicating the success attained in image editing within the field of instruction-based video editing is non-trivial. Some promising video editing solutions (Jiang et al., 2025; Hu et al., 2025) often require input masks to localize editing regions or task-specific configurations, limiting their practicality for use in the real-world. Steering video editing based on sole textual instruction is still a problem not yet fully explored in the literature.

Inspired by the success of in-context generation paradigm in image editing (Zhang et al., 2025e; Huang et al., 2024a) with both data efficiency and generation quality, we construct a joint source-target video diffusion framework for instruction-based video editing. Due to the inherent temporal complexities, two major challenges are rising when shaping in-context learning for video generation: 1) how to accurately localize the editing region when there is only text instruction? 2) how to further decrease the content interference from source editing region to the novel object generation in target video? Following the recipe for regional constraint modeling (Hu et al., 2021) in visual processing, we address the two issues by modeling the region-wise relationship on both video latents and attention maps. We mitigate the first issue through increasing the latent discrepancy in the editing region between source and target videos, and decreasing that in the non-editing areas, which enforces content regeneration in editing area with consistency of background. To alleviate the second issue, we suppress the attention of tokens in editing region to tokens in the same area of source video, alleviating the token interference from original contents. This term also encourages the novel object generation to leverage more information from tokens

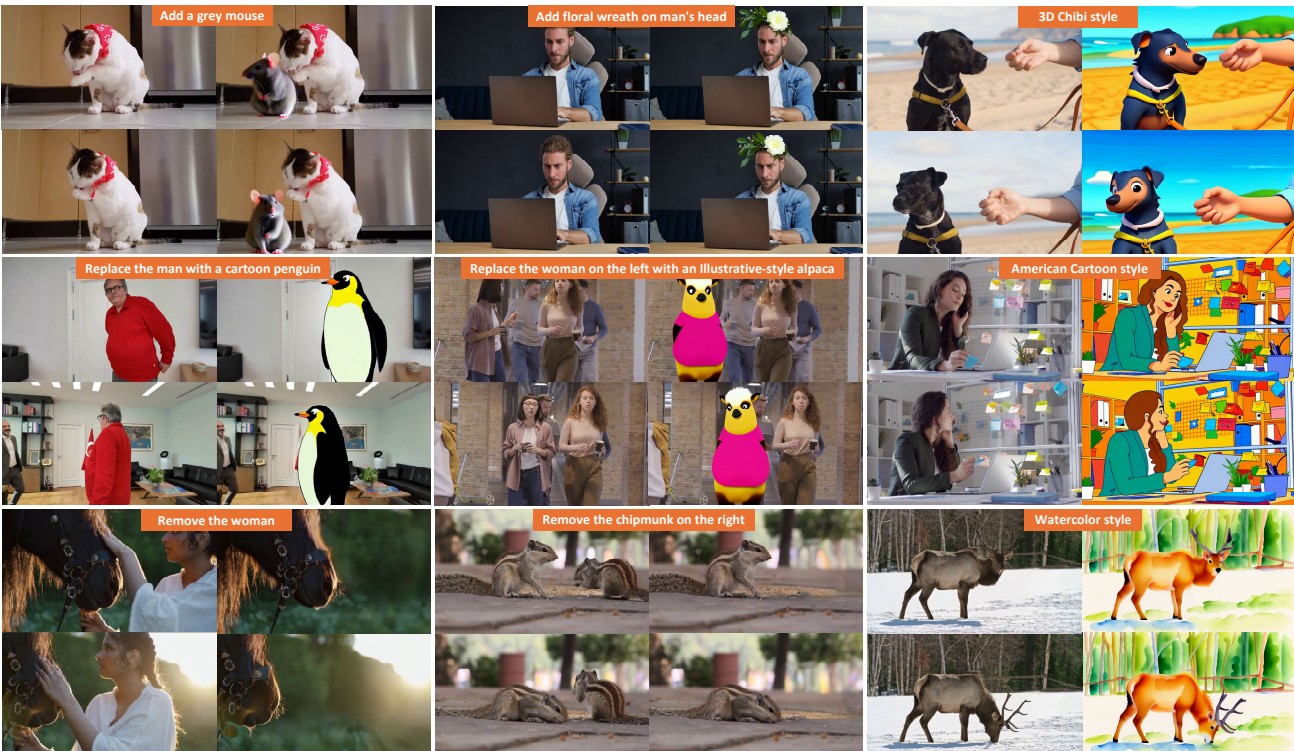

*Figure 1.* Our ReCo enables video editing based on sole textual instructions, achieving precise and high-fidelity video content modification. ReCo can adeptly handle diverse and challenging video editing tasks, including both local object editing and global style transfer.

in the background of target video itself, achieving better coherence with background.

By consolidating the idea of region-constraint in-context generation, we present a novel framework dubbed ReCo for instruction-based video editing. Technically, ReCo first concatenates the source and target videos along the left-right panel, and conducts joint video denoising for editing generation. In each training step, the paired video latents are first estimated through one-step backward diffusion process. Then, ReCo calculates latent difference between source and target video latents, and further conducts a pair-wise constraint to increase the latent discrepancy of the editing region and decrease that of non-editing areas. The similar regularization term is also performed on attention maps of DiT blocks, to suppress the concentration of tokens in the editing region on the tokens of the same region in source video. Besides, the attention of tokens in the editing region to the background of target video itself are strengthened for harmonious composition between the novel objects and background. The whole framework is jointly optimized by the flow-matching diffusion loss and the two region-constraint regularization terms.

The main contribution of this work is the new region-constraint in-context generation paradigm for instruction-based video editing. Beyond the architecture design, we meticulously construct a large-scale, high-quality video editing dataset, i.e., ReCo-Data, with 500K instruction-video pairs covering a wide spectrum of editing tasks to facilitate community research of instructional video editing. Extensive experiments further verify the effectiveness of ReCo in terms of both editing accuracy and quality.

## 2. Related Work

**Instruction-based Image Editing.** Recently, the remarkable progress achieved by text-to-image generation (Rombach et al., 2022; Lipman et al., 2023; Esser et al., 2024) encourages the development of instruction-guided image editing (Brooks et al., 2023; Hertz et al., 2023; Kulikov et al., 2025). InstructPix2Pix (Brooks et al., 2023), as one representative work in this domain, establishes a highly effective image editing data construction pipeline and achieves promising editing results. Subsequent works treat this data pipeline as a prototype, and refine it to provide more data for training powerful instruction-based editor. Based on the recipe, multi-modal models like Emu Edit (Sheynin et al., 2024), OmniGen (Xiao et al., 2025), ICEdit (Zhang et al., 2025e), HiDream-E1 (Cai et al., 2025), HiDream-O1-Image (Cai et al., 2026), Flux-Kontext (Labs et al., 2025), Qwen-Image (Wu et al., 2025a), and Nano-Banana (Anil et al., 2023) further unlock the complex capabilities, such

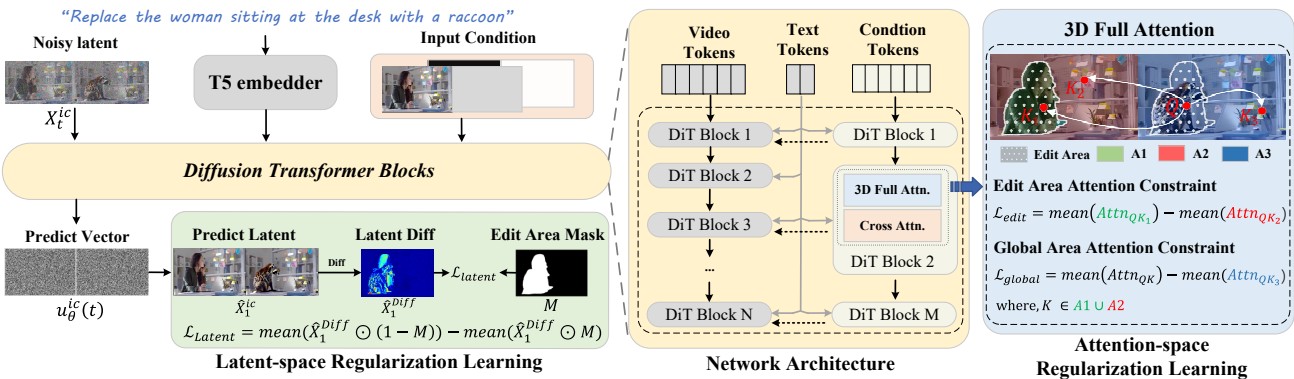

*Figure 2.* An overview of our ReCo framework. We reformulate the instructional video editing task as an *in-context generation* paradigm, guided by the source video and instruction prompt. The source video is treated as an explicit condition via feeding it into an auxiliary video condition branch. To emphasize editing modifications and alleviate the tokens interference between editing and non-editing areas, ReCo introduces two region-based constraints: (1) Latent-space regularization, which increases the latent discrepancy of the editing region between source and target videos while reducing that of non-editing areas. (2) Attention-space regularization, which suppresses the attention of the target edit region towards the corresponding region in the source video, thereby mitigating inherent token interference, while simultaneously strengthening the attention on its own generated content.

as local editing and scene transformations, even without specific fine-tuning.

Nevertheless, it is not a trivial task to replicate the success of image editing in the realm of instructional video editing. The challenge of video editing lies not only in data scarcity but also in the critical need to simultaneously handle the intricate dependencies between spatial and temporal tokens. In this work, we address the challenges through an in-context generation paradigm along with regional constraint modeling, supported by our newly constructed ReCo-Data.

**Training-free Video Editing.** Early attempts (Qi et al., 2023; Kara et al., 2024; Geyer et al., 2024; Ku et al., 2024; Liu et al., 2024b; Li et al., 2024; Cong et al., 2024) on instruction-based video editing generally leverage the training-free inference paradigm, which adapt pre-trained text-to-image diffusion models for frame-wise video editing. For instance, FateZero (Qi et al., 2023) edits video frames via DDIM (Song et al., 2021) inversion. However, the lack of temporal modeling leads to the issue of temporal inconsistency. To address this, following works such as TokenFlow (Geyer et al., 2024), VidToMe (Li et al., 2024), FLATTEN (Cong et al., 2024), and RAVE (Kara et al., 2024) employ token-merging or similarity constraints to enhance temporal coherence. There are also some approaches (e.g., FlowEdit (Kulikov et al., 2025) and FlowDirector (Li et al., 2025b)) that exploit advanced text-to-video diffusion models for more accurate diffusion inversion. Despite the flexibility of the training-free paradigm, the video quality and the model generalization ability are still the inherent limitations.

**Training-based Video Editing.** The primary obstacle for the development of training-based video editing is the profound scarcity of large-scale, high-quality paired training

data. Early approaches overcome this difficulty using one-shot tuning techniques, such as Tune-A-Video (Wu et al., 2023) and Video-P2P (Liu et al., 2024b), but still struggle to achieve ideal editing results. In another direction, several works (Cheng et al., 2024; Liu et al., 2025b; Wu et al., 2025b; Zhang et al., 2025c) push the boundary of video editing in terms of both dataset construction and framework design. For example, GenProp (Liu et al., 2025b), Senorita (Zi et al., 2025), and FFP-300K (Huang et al., 2026) formulate video editing as an image-propagation process, in which the first frame is edited first and the content modification is then propagated to subsequent frames. More recently, Lucy-Edit (DecartAI Team, 2025) and Ditto (Bai et al., 2025a) propose to train video editing models directly on source-target videos and text instructions. Lucy-Edit concatenates source video latents with denoised video latents as the condition, while Ditto learns the condition through a ControlNet manner. However, these methods struggle to capture the semantics of source video, leading to inaccurate editing. Inspired by in-context learning (Zhang et al., 2025e), we propose ReCo, which promotes a deeper semantic alignment by jointly generating the source and the target, thereby ensuring highly accurate and consistent video editing.

## 3. Our Approach

Here we will introduce ReCo, a novel region-constrained in-context video generation framework for instructional video editing. The overall architecture is illustrated in Figure 2. Given a pair of source and target videos, ReCo reformulates the generation process into an in-context learning paradigm, achieved by width-wise concatenating the two videos for joint denoising. Simultaneously, to ensure the faithful preservation of source video information, we employ an additional video condition branch that explores

the condition learning on the source video. In the training stage, we introduce two regularization terms, i.e., latent and attention regularization, to benefit accurate video editing learning without pre-specified editing regions. The latent regularization learning are conducted on the one-step backward denoised latents to amplify region-wise modifications and the consistency of background. Meanwhile, the attention regularization term suppresses the attention of newly generated objects of the target video on the source video's editing region, thereby decreasing the token interference from the original visual content.

## 3.1. Preliminaries: Video DiT Training

To leverage the prior knowledge from pre-trained video generation models (Kong et al., 2024; Wan et al., 2025; Ha-Cohen et al., 2025; Zhang et al., 2025b; 2024), we adopt an advanced video diffusion transformer, i.e., Wan-T2V-1.3B (Wan et al., 2025), as the backbone architecture for ReCo. To facilitate a clear understanding of our proposal, we first review the training procedure of video DiT. Typically, most video DiT models are grounded in flow matching (Lipman et al., 2023; Esser et al., 2024) theory, which provides a theoretically rigorous framework for learning continuous-time generative processes. It aims to learn a vector field that smoothly transports samples from a simple prior distribution $P_0$ (e.g., a Gaussian $\mathcal{N}(0, 1)$) to the target data distribution $P_1$.

Given the video latent $x_1$ in training, a random noise sample $x_0 \sim \mathcal{N}(0, 1)$ and a timestep $t \in [0, 1]$ are sampled from a logit-normal distribution. Then, $x_0$ is combined with $x_1$ to obtain an intermediate noised latent $x_t$ via the forward diffusion process based on Rectified Flow (Esser et al., 2024):

$$x_t = tx_1 + (1 - t)x_0. \tag{1}$$

Then, the ground-truth velocity vector is calculated as:

$$v_t = \frac{dx_t}{dt} = x_1 - x_0. \tag{2}$$

The video DiT model is learned to estimate this vector via:

$$u_\theta(t) = u(x_t, c, t; \theta), \tag{3}$$

where $x_t$ is the noisy latent, $\theta$ represents the model parameters, and $c$ is the set of input conditions. For the instructional video editing task, $c$ comprises both the textual instruction and the source video. Therefore, the training objective $\mathcal{L}$ is defined as the mean squared error (MSE) between the model's output and the ground-truth velocity $v_t$:

$$\mathcal{L} = \mathbb{E}_{x_0, x_1, c, t} \left\| u(x_t, c, t; \theta) - v_t \right\|^2. \tag{4}$$

The objective illustrates that the target vector at any given timestep $t$ (i.e., the instantaneous velocity) is simply formulated as $x_1 - x_0$. The target is exceptionally clear and stable,

making it straightforward for the neural network to learn, which in turn yields high-quality video generation.

## 3.2. In-Context Generation for Video Editing

The in-context generation paradigm has recently demonstrated significant advantages in image editing (Zhang et al., 2025e; Tan et al., 2025; Shin et al., 2025; Huang et al., 2024a), particularly in terms of data efficiency and generation quality. Inspired by this, we reformulate the video editing process as in-context generation. Technically, given the video latent pair (i.e., the source video $x_1^{src}$ and the target video $x_1^{tar}$), we *width-wise* concatenate them to form a single in-context video latent $x_1^{ic}$ as:

$$x_1^{ic} = [x_1^{src}, x_1^{tar}]. \tag{5}$$

During model training, a noise latent $x_0^{ic}$ is sampled from a Gaussian distribution and then added to corrupt $x_1^{ic}$, producing the noisy latent $x_t^{ic}$ which is fed into video DiT for joint source and target video denoising:

$$x_t^{ic} = tx_1^{ic} + (1 - t)x_0^{ic}. \tag{6}$$

The ground-truth velocity vector is reformulated as:

$$v_t^{ic} = \frac{dx_t^{ic}}{dt} = x_1^{ic} - x_0^{ic}. \tag{7}$$

Consequently, we adapt the training objective Eq.(4) to the in-context generation paradigm and form $\mathcal{L}_{ic}$ as:

$$u_\theta^{ic}(t) = u(x_t^{ic}, c, t; \theta), \tag{8}$$

$$\mathcal{L}_{ic} = \mathbb{E}_{x_0^{ic}, x_1^{ic}, c, t} \left\| u(x_t^{ic}, c, t; \theta) - v_t^{ic} \right\|^2. \tag{9}$$

In our in-context generation scenario, the predicted vector $u_\theta^{ic}(t)$ is required to jointly learn both the reconstruction of the source video and the generation of the edited video. Due to the high correlation between the source and target videos in editing, such joint learning facilitates strong token interaction, leading to superior video editing performance. Simultaneously, we employ a video condition branch, as depicted in Figure 2, to ensure that the video condition is comprehensively learned to calibrate video denoising. Moreover, we exploit the Low-Rank Adaptation (LoRA) technique for efficient and stable video DiT fine-tuning.

## 3.3. Regional Constraint in Latent Space

The in-context generation benefits token interaction between source and target videos for better instructional video editing. However, compared to prior advances that require pre-specified edit regions (Jiang et al., 2025; Bian et al., 2025; Hu et al., 2025) in video editing, solely relying on textual instruction might still lead to the issue of inaccurate

editing region. To alleviate this limitation, we introduce a regional constraint within the latent space. This mechanism is designed to increase the latent discrepancy of the editing region between source and target videos while reducing that of non-editing areas, amplifying the modification on editing area and alleviating outside unexpected content generation, respectively.

Given the velocity vector $u_\theta^{ic}(t)$ estimated by video DiT with timestep $t$, we first derive the one-step backward denoised latent $\hat{x}_1^{ic}$ based on the Rectified Flow definition as:

$$\hat{x}_1^{ic} = x_t^{ic} + (1-t)u_\theta^{ic}(t). \tag{10}$$

The obtained denoised video latent $\hat{x}_1^{ic}$ is then divided along width dimension to get its source and target parts as follows:

$$[\hat{x}_1^{src}, \hat{x}_1^{tar}] = \hat{x}_1^{ic}. \tag{11}$$

Next, we calculate the latent difference vector $\hat{X}_1^{Diff}$ between $\hat{x}_1^{src}$ and $\hat{x}_1^{tar}$ through:

$$\hat{X}_1^{Diff} = \left|\hat{x}_1^{tar} - \hat{x}_1^{src}\right|. \tag{12}$$

For a successful editing, we hypothesize that the latent discrepancy should be high within the editing region between the source and target videos, while the non-editing regions should remain unchanged. To achieve this, we introduce the latent-space regional constraint $\mathcal{L}_{\text{latent}}$ to regulate DiT training. Let $M$ be the binary latent mask indicating the editing region (where $M = 1$ denotes regions that should be edited). $\mathcal{L}_{\text{latent}}$ aims to minimize the mean discrepancy in the non-editing regions while maximizing it within the editing regions, which is computed by:

$$\begin{aligned}\mathcal{L}_{\text{latent}} = \ &\text{mean}\left(\hat{X}_1^{Diff} \odot (1-M)\right) \\ &- \text{mean}\left(\hat{X}_1^{Diff} \odot M\right).\end{aligned} \tag{13}$$

### 3.4. Regional Constraint in Attention Space

Besides the region constraint on latent space, the robust learning of in-context generation also necessitates alleviating inherent token interference between the editing and non-editing regions at attention level. For instance, there should be less concentration on the original contents of editing region in source video, and more attention on its own generated background for better coherence. To formulate these relative correlations on attention, we propose to regulate the attention map learning from two perspectives, i.e., the relative relationship within editing region, and the relative relationship within the entire video regions.

As shown in the right part of Figure 2, we first partition the whole area of source-target video pair into three distinct key regions: the source video's editing region (A1), the source

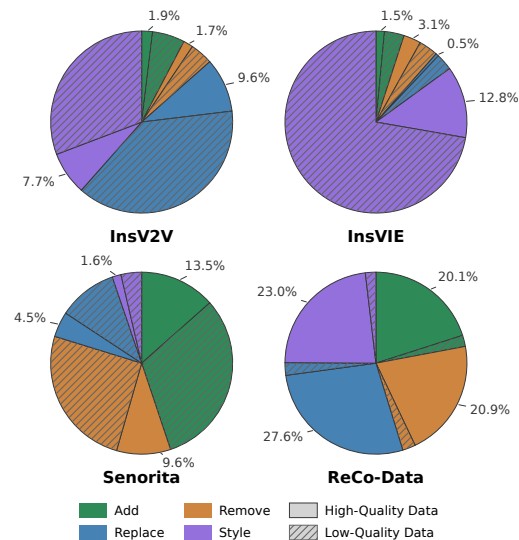

*Figure 3.* Comparison between existing video editing datasets and our ReCo-Data. Ours features the most balanced data distribution and has a higher ratio of the high-quality samples.

video's non-editing region (A2), and the entire target video region (A3). To formulate the relative relationship within editing region for attention learning, tokens from the target editing region (queries $Q$) should reduce their attention to the corresponding source editing region (keys $K_1$). We define this as the *edit attention loss* $\mathcal{L}_{edit}$:

$$\mathcal{L}_{edit} = \text{mean}(Attn_{QK_1}) - \text{mean}(Attn_{QK_2}), \tag{14}$$

where $Attn_{QK}$ is the similarity score between queries $Q$ and keys $K$. Furthermore, to guarantee coherent integration of generated content with the background, the queries $Q$ should reduce the overall reliance on the entire source video (e.g., keys $K$ in $A1 \cup A2$), while focusing more on the contextually relevant of target video regions (e.g., keys $K_3$ in $A3$). Therefore, such type of constraint is formulated as the *global attention loss* $\mathcal{L}_{global}$:

$$\mathcal{L}_{global} = \text{mean}(Attn_{QK}) - \text{mean}(Attn_{QK_3}), \tag{15}$$

The attention-space regional constraint is thus defined as the sum of both two components:

$$\mathcal{L}_{attn} = \mathcal{L}_{edit} + \mathcal{L}_{global}. \tag{16}$$

Finally, the overall training objective in our ReCo is formulated as a multi-task loss by integrating basic in-context flow matching loss $\mathcal{L}_{ic}$ and two regional constraints in both latent space $\mathcal{L}_{\text{latent}}$ and attention space $\mathcal{L}_{attn}$:

$$\mathcal{L} = \mathcal{L}_{ic} + \lambda_1 \mathcal{L}_{\text{latent}} + \lambda_2 \mathcal{L}_{attn}, \tag{17}$$

where $\lambda_1$ and $\lambda_2$ are trade-off parameters. The two constraints emphasize more accurate editing regions and the learning of correct token relationships, mitigating token interference for more natural video content generation.

| Source Video | InsViE | Lucy-Edit | Ditto | ReCo |
|---|---|---|---|---|

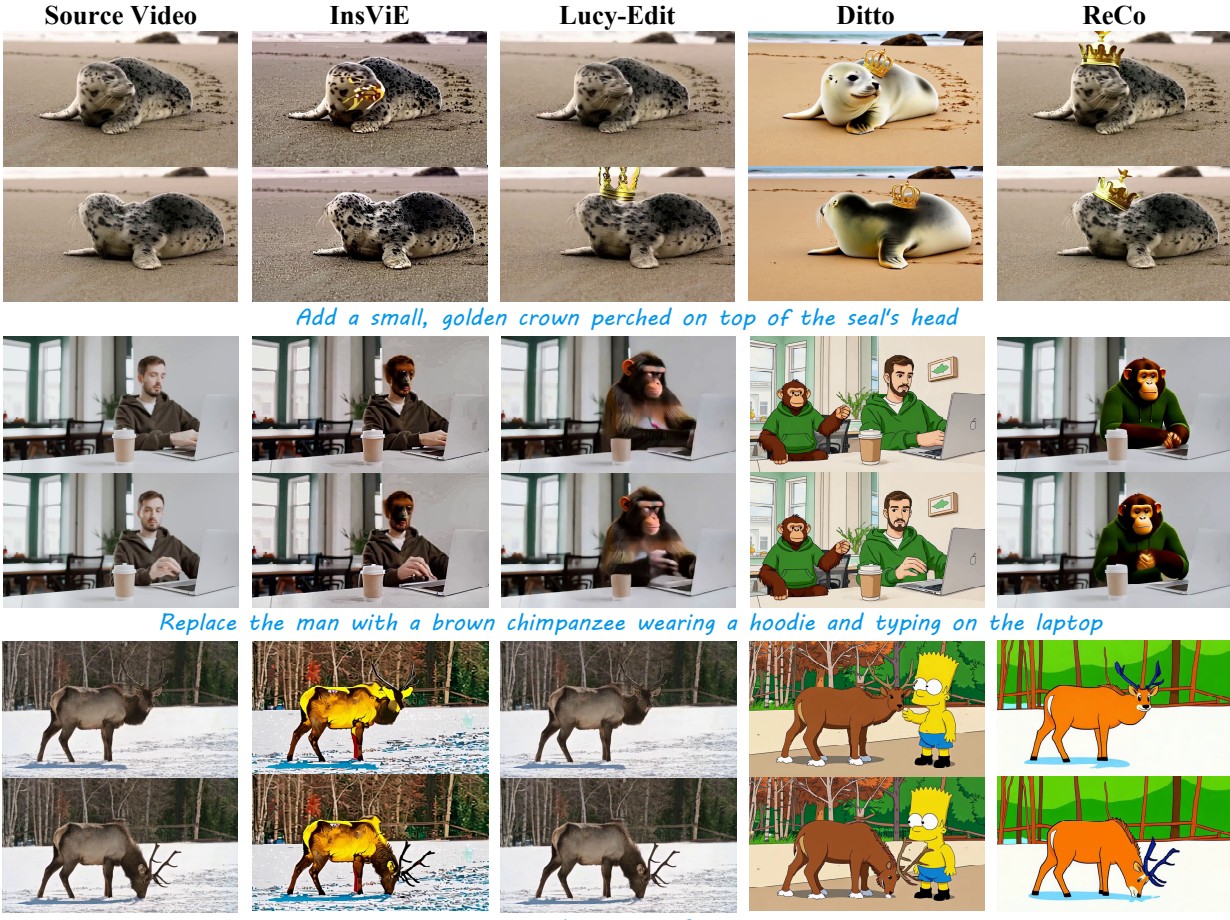

*Add a small, golden crown perched on top of the seal's head*

*Replace the man with a brown chimpanzee wearing a hoodie and typing on the laptop*

*Convert the video into a Simpsons style*

*Figure 4.* Examples of video editing (i.e., add object, replace object and style transfer) results by different approaches.

## 4. Experiments

### 4.1. Experimental Settings

**Datasets.** Despite recent great progress in instructional video editing, a significant bottleneck still remains: the lack of a large-scale, high-quality training dataset. To address this challenge, we introduce the **ReCo-Data**, which is meticulously curated to support four major video editing tasks: instance-level object adding, removing, and replacing, and the global video stylization. Our data construction pipeline involves six main stages: (1) raw data pre-process; (2) object segmentation; (3) instruction generation using VLLMs (i.e., Gemini-2.5-Flash-Thinking (Anil et al., 2023)); (4) condition pairs construction; (5) video synthesis using VACE (Jiang et al., 2025); and (6) video filtering and re-captioning with VLLMs. More details are provided in the Appendix A. Ultimately, we construct ReCo-Data with 500K high-quality instruction-video pairs. Each video clip contains 81 frames with the resolution of $480 \times 832$. The video duration is $5.0$ seconds.

We compare ReCo-Data with existing video editing datasets in terms of the ratio of high-quality samples, which reflects the usability and overall quality of the dataset. Specifically, we randomly sample 200 video editing pairs from each editing task across all datasets, and invite 10 evaluators to qualitatively assess the video editing quality. As shown in Figure 3, the ratio of high-quality samples in existing datasets (i.e., InsV2V (Cheng et al., 2024), InsVIE (Wu et al., 2025b), and Senorita (Zi et al., 2025)) is usually low ($17.9\% \sim 29.2\%$). It indicates that these datasets have not undergone rigorous data cleaning processes, and the large number of low-quality samples makes them suboptimal for training high-performing instructional video editing models. Besides, the cost of data re-cleaning is extremely high, while the potential benefit is minimal due to the low frame rate, low resolution, and poor synthesis quality of previous datasets. Instead, our ReCo-Data has a very high proportion ($91.6\%$) of high-quality samples and a well-balanced data distribution across different tasks. It can be readily used for model training without any pre-processing. The usability of ReCo-Data is also verified by the training of our model.

**Benchmarks.** We construct a video editing evaluation benchmark which contains 480 video-instruction pairs, 120 pairs for each of the four video editing tasks. Since tradi-

*Table 1.* Performance comparisons on four video editing tasks (i.e., add object, replace object, remove object and style transfer).

| Task | Approach | Edit Accuracy (EA) | | | Video Naturalness (VN) | | | Video Quality (VQ) | | | Average Score | | | |
|---|---|---|---|---|---|---|---|---|---|---|---|---|---|---|
| | | SA | SP | CP | AN | SN | MN | VF | TS | ES | $S_{EA}$ | $S_{VN}$ | $S_{VQ}$ | $S$ |
| **Add** | InsViE | 2.60 | 2.79 | 2.78 | 2.33 | 3.98 | 3.74 | 3.71 | 3.91 | 3.58 | 2.60 | 3.10 | 3.46 | 3.05 |
| | Lucy-Edit | 6.27 | 6.32 | 7.75 | 4.63 | 7.08 | 6.08 | 6.31 | 6.82 | 7.57 | 6.47 | 5.70 | 6.77 | 6.31 |
| | Ditto | 7.46 | 7.24 | 6.30 | 6.30 | **8.85** | **8.30** | **8.13** | 8.55 | 9.03 | 6.70 | **7.57** | 8.41 | 7.56 |
| | ReCo | **8.65** | **8.40** | **9.22** | 6.39 | 8.78 | 8.28 | 8.02 | **8.61** | **9.61** | **8.54** | 7.55 | **8.61** | **8.23** |
| **Replace** | InsViE | 1.89 | 2.38 | 2.48 | 2.58 | 5.25 | 5.05 | 3.76 | 4.00 | 3.52 | 2.10 | 3.91 | 3.49 | 3.17 |
| | Lucy-Edit | 6.57 | 7.49 | 7.73 | 5.13 | 7.46 | 6.65 | 6.32 | 6.64 | 8.08 | 7.08 | 6.21 | 6.88 | 6.72 |
| | Ditto | 4.95 | 4.83 | 4.79 | 5.81 | 8.63 | 8.10 | 7.55 | 7.95 | 8.71 | 4.56 | 7.21 | 7.96 | 6.58 |
| | ReCo | 9.38 | 9.43 | 9.59 | 7.07 | 8.87 | 8.47 | 8.19 | 8.65 | 9.67 | 9.43 | 8.01 | 8.77 | 8.74 |
| **Remove** | InsViE | 2.53 | 2.49 | 2.44 | 2.63 | 4.87 | 4.72 | 3.41 | 3.67 | 3.40 | 2.44 | 3.76 | 3.29 | 3.16 |
| | VACE | 4.58 | 4.58 | 4.56 | 4.96 | 6.09 | 5.89 | 5.48 | 5.50 | 5.57 | 4.57 | 5.43 | 5.56 | 5.19 |
| | ReCo | 7.43 | 7.43 | 7.17 | 6.20 | 7.43 | 7.30 | 6.48 | 6.63 | 7.68 | 7.28 | 6.90 | 6.82 | 7.00 |
| **Style** | InsViE | 7.59 | 8.86 | 8.49 | 6.77 | 9.14 | 9.28 | 7.13 | 6.40 | 8.99 | 8.17 | 8.21 | 7.35 | 7.91 |
| | Lucy-Edit | 3.73 | 5.59 | 5.39 | 4.20 | 5.88 | 5.88 | 4.44 | 4.17 | 5.87 | 4.65 | 4.67 | 5.17 | 4.83 |
| | Ditto | 9.10 | 9.36 | 9.26 | 8.25 | 9.51 | 9.58 | 8.33 | 8.33 | 9.77 | 9.20 | 9.07 | 8.77 | 9.01 |
| | ReCo | **9.11** | **9.82** | **9.54** | **8.43** | **9.55** | **9.70** | **8.61** | **8.35** | **9.87** | **9.42** | **9.19** | **8.90** | **9.17** |

tional metrics usually struggle to accurately and comprehensively evaluate video editing across various dimensions, we follow the image editing advance (Wei et al., 2025), and employ a VLLM as the referee for evaluation. Considering the inherent complexity of video data, we extended the image editing metrics (Wei et al., 2025) and construct a diverse set of evaluation dimensions tailored for video editing. We measure the video editing from three main aspects: (1) *Edit Accuracy*, with the sub-dimensions of Semantic Accuracy (SA), Scope Precision (SP), and Content Preservation (CP); (2) *Video Naturalness*, which includes Appearance Naturalness (AN), Scale Naturalness (SN), and Motion Naturalness (MN); and (3) *Video Quality*, with the dimensions of Visual Fidelity (VF), Temporal Stability (TS), and Edit Stability (ES). We obtain the per-category scores (i.e., $S_{EA}$, $S_{VN}$, $S_{VQ}$) by calculating the geometric mean of all sub-dimension scores of each major perspective. The overall averaged score ($S$) is the arithmetic mean of the three per-category scores. Specifically, we feed the source and generated video pair, and the predefined system instructions into Gemini-2.5-Flash (Anil et al., 2023), and ask it to give the rating for video editing from all the nine sub-dimensions. More details of benchmark construction and the full evaluation protocol are provided in the Appendix B.

**Implementation Details.** In ReCo, we employ Wan-T2V-1.3B (Wan et al., 2025) as our base architecture. Each training sample is an 81-frame video clip, with the frame rate of 16 fps and the resolution of $480 \times 832$. For mask generation to align the resolution of video latents, we first encode the editing mask via VAE (Kingma & Welling, 2013) and then apply $k$-means clustering to binarize them. We set the rank of the LoRA as 128. ReCo is trained using the AdamW optimizer with a two-stage learning rate schedule: the model is first trained with a learning rate of $1 \times 10^{-4}$ to achieve stable convergence, followed by a fine-tuning

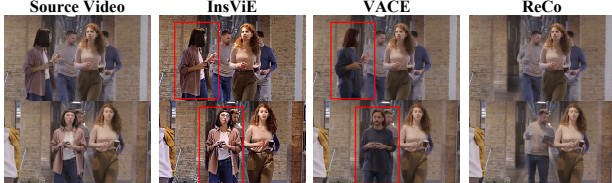

Source Video     InsViE     VACE     ReCo

*Remove the woman with glasses on the left*

*Figure 5.* Visual comparisons on the object removal task.

stage using a lower learning rate of $2 \times 10^{-5}$ for further refinement. All experiments are conducted on 24 NVIDIA A800 GPUs with a mini-batch size of 24.

### 4.2. Comparisons with State-of-the-Art Methods

We compare our ReCo with several state-of-the-art instructional video editing methods, including InsViE (Wu et al., 2025b), Ditto (Bai et al., 2025a), Lucy-Edit (DecartAI Team, 2025), and VACE (Jiang et al., 2025), on our VLLM-based benchmark. Table 1 summarizes the performance comparisons on the four video editing tasks. Overall, ReCo consistently outperforms existing baselines on the total score $S$ across all tasks. In particular, for the local video editing, ReCo attains the total score $S$ of 8.23 on *Add* and 8.74 on *Replace*, surpassing the strong competitor Ditto (7.56) and Lucy-Edit (6.72) by 0.67 and 2.02, respectively. Significant performance trends can also be observed on the editing accuracy perspective (i.e., $S_{EA}$). The results demonstrate that our ReCo not only accurately follows the instruction prompt to correctly localize the editing region but also preserves the contents of non-edited areas. In terms of video naturalness (i.e., $S_{VN}$), the better performances achieved by our model verifies the efficacy of naturally integrate editing objects into source video. Although the $S_{VN}$ of ReCo is slightly below that of Ditto on the *Add* task, ours can better keep original video contents while Ditto tends to re-render the

*Table 2.* Performance comparisons among different variants of ReCo on four video editing tasks.

| Model | Add | | | | Replace | | | | Remove | | | | Style | | | |
|---|---|---|---|---|---|---|---|---|---|---|---|---|---|---|---|---|
| | $S_{EA}$ | $S_{VN}$ | $S_{VQ}$ | $S$ | $S_{EA}$ | $S_{VN}$ | $S_{VQ}$ | $S$ | $S_{EA}$ | $S_{VN}$ | $S_{VQ}$ | $S$ | $S_{EA}$ | $S_{VN}$ | $S_{VQ}$ | $S$ |
| ReCo$_{LC-}$ | 8.05 | 7.44 | 8.59 | 8.03 | 9.01 | 8.01 | 8.67 | 8.56 | 6.90 | 6.83 | **6.91** | 6.88 | 9.09 | 9.10 | 8.84 | 9.01 |
| ReCo$_{AC-}$ | 8.33 | 7.37 | 8.01 | 7.90 | 9.23 | 7.94 | 8.46 | 8.54 | 7.11 | 6.75 | 6.70 | 6.85 | 9.21 | 9.08 | 8.81 | 9.03 |
| ReCo | **8.54** | **7.55** | **8.61** | **8.23** | **9.43** | **8.01** | **8.77** | **8.74** | **7.28** | **6.90** | 6.82 | **7.00** | **9.43** | **9.19** | **8.90** | **9.17** |

whole video into different color style as shown in Figure 4. The phenomenon is also evidenced by the lower $S_{EA}$ score (6.70) of Ditto. Additionally, the best performance of video quality ($S_{VQ}$) further indicates that the videos generated by ReCo have minimal visual artifacts or degradation. Even under the multi-task training setting (i.e., unify local editing and global stylization) that could bring some conflicts during model optimization, ReCo still manifests the strong capability for video style transfer and attains 9.17 of the total score $S$. All these results basically validate the merit of performing regional constraint modeling on in-context generation for instructional video editing.

Figure 4 and 5 further show the video editing results on the four tasks. Generally, compared to other baselines, ReCo edits videos with better instruction following, higher video quality and better background consistency. For instance, InsViE tends to produce videos with artifacts and usually suffers from editing failure. Recent Lucy-Edit exhibits poor instruction-following and fails to accurately render the specified attributes (e.g., brown chimpanzee wearing a hoodie). Though Ditto generates natural-looking objects in the *Add* task, it struggles to preserve background consistency of non-editing regions and localize the accurate editing region (e.g., adding the crown at the back of the seal). Meanwhile, the ability of instruction following for Ditto is inferior to ours, erroneously synthesizing a new monkey alongside the man instead of replacing him. We speculate that Ditto's ControlNet-based conditioning struggles to capture source semantics, leading to misaligned edits. Conversely, ReCo's in-context generation fosters deeper semantic interaction by jointly generating the source and the target. By further incorporating region-wise constraints, ReCo ensures precise localization, while effectively mitigating cross-region token interference. Thus, the videos modified by ReCo reflect both accurate editing results and natural novel object integration with the original video background.

### 4.3. Ablation Study on Regional Constraint

We investigate how the two regional constraints in our ReCo influence the final instruction-based video editing. Table 2 summarizes the video editing performances of different variants of our ReCo. Two additional runs are involved, i.e., ReCo$_{LC-}$ and ReCo$_{AC-}$, which remove the latent and attention regional constraint in ReCo, respectively. Specifically, when the region constraint in latent space is discarded, there is a dramatic performance drop on $S_{EA}$, which indi-

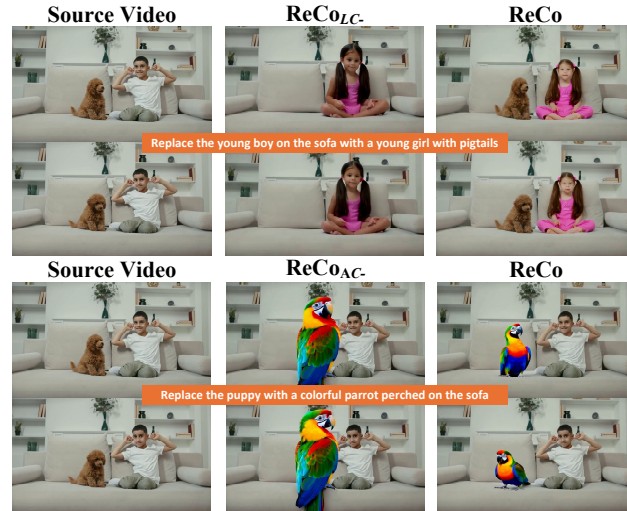

*Figure 6.* Editing results on replace task among variants of ReCo.

cates a significant decay of editing accuracy. The scores of $S_{VN}$ and $S_{VQ}$ also decrease slightly but remain comparable. The results highlight the effectiveness of latent region constraint learning to amplify accurate localization of editing region. The top part of Figure 6 further visualizes one video editing case among ReCo$_{LC-}$ and ReCo. Given the instruction of "replace the young boy on the sofa with a young girl with pigtails," ReCo$_{LC-}$ could replace the boy but incorrectly removes the nearby dog.

When removing the regularization term in attention space, ReCo$_{AC-}$ performs worse on the video naturalness perspective (i.e., $S_{VN}$) as shown in Table 2. We also show one editing example in the lower part of Figure 6. As shown in the figure, ReCo$_{AC-}$ generates a big parrot which has an unnatural scale relative to the environment. With the equipment of attention regularization that reduces the token interference from editing area and strengthens the interaction with background in novel object generation, ReCo synthesizes the parrot with natural size and better coherence.

### 4.4. Ablation Study on Conditioning Strategy

To verify the advantages of in-context learning, we designed a baseline (i.e., Wan+ControlNet) that learns video editing in a ControlNet manner. To adapt the model to instructional prompts, we also applied LoRA fine-tuning to the DiT backbone with the same hyper-parameters as ReCo.

As shown in Table 3, the editing quality of ControlNet

*Table 3.* Evaluation on the effectiveness of in-context generation paradigm. We compare ReCo against a ControlNet-based baseline to demonstrate the advantages of the in-context generation for instructional video editing.

| Model | Add | | | | Replace | | | | Remove | | | | Style | | | |
|---|---|---|---|---|---|---|---|---|---|---|---|---|---|---|---|---|
| | $S_{EA}$ | $S_{VN}$ | $S_{VQ}$ | $S$ | $S_{EA}$ | $S_{VN}$ | $S_{VQ}$ | $S$ | $S_{EA}$ | $S_{VN}$ | $S_{VQ}$ | $S$ | $S_{EA}$ | $S_{VN}$ | $S_{VQ}$ | $S$ |
| Wan+ControlNet | 7.65 | **7.86** | 7.99 | 7.83 | 8.01 | **8.46** | 7.74 | 8.07 | 7.05 | 6.38 | 6.32 | 6.58 | 8.72 | 8.99 | **9.09** | 8.93 |
| ReCo | **8.54** | 7.55 | **8.61** | **8.23** | **9.43** | 8.01 | **8.77** | **8.74** | **7.28** | **6.90** | **6.82** | **7.00** | **9.43** | **9.19** | 8.90 | **9.17** |

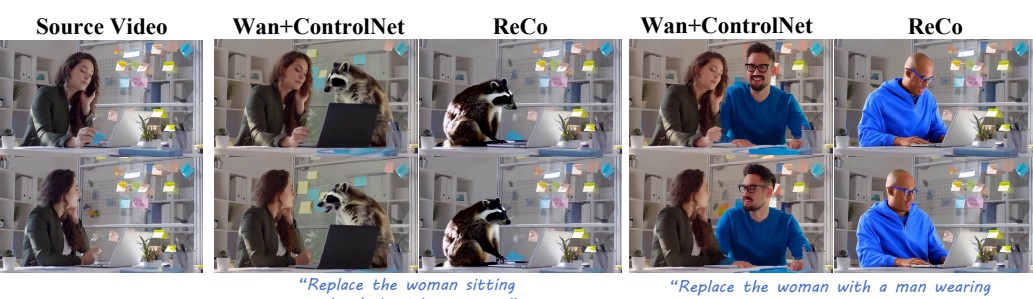

Source Video     Wan+ControlNet     ReCo     Wan+ControlNet     ReCo

*"Replace the woman sitting at the desk with a raccoon"*     *"Replace the woman with a man wearing a blue shirt and glasses"*

*Figure 7.* Visual comparison of object replacement results between the ControlNet-based baseline and our ReCo.

model degrades significantly. For instance, the $S_{EA}$ score drops by nearly $1.42$ on object replacement task. Although ControlNet effectively preserves the structural and textural information of the video, the model struggles to comprehend the underlying semantic content. Consequently, it fails to execute precise edits based on short instructions, leading to the random insertion of novel objects. As illustrated in Figure 7, the baseline merely places a raccoon into the scene without performing replacement. In contrast, our in-context learning paradigm requires the model to simultaneously generate both the original and edited videos. Such dual reconstruction process facilitates a deeper semantic understanding of the source video, enabling the model to accurately synthesize the intended edits.

Additionally, when the ControlNet model attempts to handle four distinct editing tasks simultaneously, the model can suffer from poor convergence. Since the task like "Stylization" requires global modifications while others require local content changes, a lack of semantic understanding could lead the model to simply mimic data distributions. This will result in unintended shifts in the overall style and color tone. The same issue can also occur on local replacement task, consequently causing a drastic reduction in $S_{EA}$. Instead, the in-context generation of our ReCo empowers the model to understand short instructions effectively across all four tasks. It benefits the high-quality stylization while maintaining the integrity of local edits without distorting the global appearance of source video.

## 5. Conclusions

We have presented ReCo that shapes in-context generation for instruction-based video editing. Particularly, we study the problem of integrating the regional constraint modeling between editing and non-editing areas into diffusion training. To materialize our idea, ReCo jointly denoises the width-concatenated source-target video pair based on the natural language instructions, and conducts two regularization terms to emphasize region-wise relationship on both one-step backward denoised latents and attention maps. To alleviate unexpected content generation in non-editing regions, the regularization term in latent space tries to decrease the latent discrepancy of non-editing regions between source and target videos, while increasing the differences at the editing area. Meanwhile, ReCo suppresses the attention of tokens in the editing region to tokens in the same part of source video, which alleviates the interference from original editing region tokens to novel object generation. Moreover, we carefully construct a high-quality video editing dataset, i.e., ReCo-Data, consisting of 500K instruction-video pairs covering a wide range of editing tasks. Extensive experiments across four editing tasks verify the superiority of ReCo over state-of-the-art approaches.

## Impact Statement

This paper aims to advance the field of instructional video editing. As a video generative model, our approach enables local object editing and global style transfer of videos, with prospective applications in multimedia creation. Although video editing models could potentially enable misuse, existing content detection techniques can help mitigate such risks. Furthermore, we have emphasized the importance of ethical considerations in both stages of ReCo-Data construction and ReCo model training.

## Acknowledgements

This work was supported by the Key Science and Technology Project of Anhui Province under Grant No. 202523o09050002 and the National Nature Science Foundation of China under Grant U24A20329.

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

The appendix contains: A). the construction pipeline of ReCo-Data; B). the details of VLLM-based benchmark for evaluation; C). the implementation details of baselines and ReCo; D). more evaluations on ReCo; E). additional analysis and qualitative results of ReCo.

## A. Construction Pipeline of ReCo-Data

Though instructional video editing has seen remarkable advances recently, the absence of large-scale, high-quality training datasets remains a critical hurdle. To overcome this, we present **ReCo-Data**, a dataset carefully designed to facilitate four key editing tasks: instance-level object addition, removal, replacement, and global video stylization.

As illustrated in Figure 8, the construction pipeline of ReCo-Data consists of six primary stages: (1) raw data pre-processing, where we filter raw video data based on specific quality criteria; (2) object segmentation, extracting object mask from videos; (3) instruction generation, employing VLLM (i.e., Gemini-2.5-Flash-Thinking (Anil et al., 2023)) to construct editing prompts; (4) condition pair construction, which involves first frame editing and depth map generation to prepare the input conditions for VACE (Jiang et al., 2025); (5) video synthesis, employing VACE to generate videos based on conditions; and (6) video filtering and re-captioning, where VLLM (i.e., Gemini-2.5-Flash-Thinking (Anil et al., 2023)) is leveraged again to filter out low-quality samples and re-caption remained videos.

### A.1. Raw Data Pre-processing

**Data Collection.** To ensure the data diversity, we collect raw videos from two distinct sources: the HD-VG (Wang et al., 2023) dataset and videos from Pixel website (Bian et al., 2025). We employ PySceneDetect (PySceneDetect Developers, 2024) to segment the long, multi-scene videos into shorter, manageable clips.

**Data Filtering.** We first filter clips based on basic metadata, retaining those with a duration exceeding 5 seconds, a frame rate greater than 24 fps, and a resolution of at least 720P. Then, we utilize aesthetic scores (christophschuhmann, 2024) and optical flow (Teed & Deng, 2020) to select videos characterized by high aesthetic quality and appropriate motion magnitude. Finally, to ensure visual purity, we employ PaddleOCR (Cui et al., 2025) for watermark detection, spatially cropping the frames to exclude any text detected with a confidence score exceeding 0.7.

**Video Captioning.** For subsequent object segmentation and editing prompt construction, we utilize Qwen2.5-VL-32B (Bai et al., 2025c) to obtain detailed descriptions of remained videos.

### A.2. Object Segmentation

To enable precise instance-level object editing (e.g., replacement and removal) by using video inpainting models like VACE (Jiang et al., 2025), we need to first isolate the target objects. Given the complexity of scenes containing multiple objects, we adopt a systematic segmentation approach. First, we define a taxonomy and employ a Named Entity Recognition (NER) model, i.e., SpaCy (spaCy Developers, 2024), to extract relevant entity nouns from video captions. Subsequently, we utilize Grounding Dino (Ren et al., 2024) to detect objects and obtain their bounding boxes. To ensure the quality of the proposals, we apply Non-Maximum Suppression (NMS) to filter out duplicate boxes with overlaps exceeding 25% and discard boxes that are disproportionately large or small. Finally, using the bounding boxes as prompts, we employ SAM 2 (Nikhila et al., 2025) to generate mask sequences for the target objects.

### A.3. Instruction Generation

The protocols to prepare editing prompts exhibit variations across local editing and global video stylization tasks.

**Local Editing.** For the local video editing tasks, we provide Gemini with a tuple containing original video caption and one representative key frame. In this key frame, the target object region is explicitly highlighted with a red convex hull. Guided by a finely tuned system prompt, Gemini is required to generate an appropriate editing instruction along with a target video caption describing the post-edit state.

**Video Stylization.** The process is analogous to local editing. We leverage Gemini's creative capabilities to brainstorm diverse stylization instructions and generate the corresponding target video descriptions.

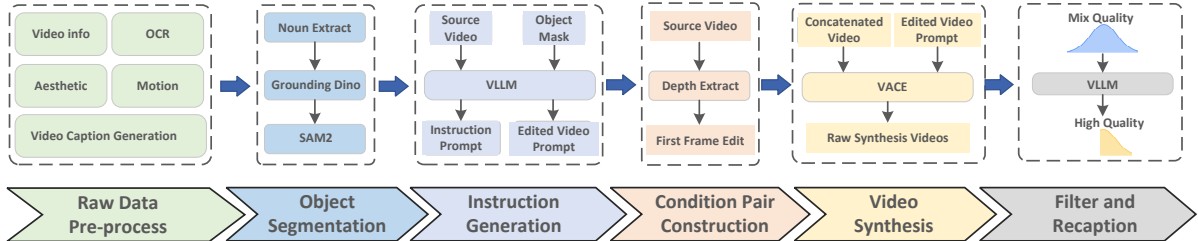

*Figure 8.* An overview of our data construction pipeline. The process consists of six main stages: raw data pre-processing, object segmentation, instruction generation, condition pair construction, video synthesis, and video filtering and re-captioning.

## A.4. Condition Pair Construction

In this stage, we leverage the full capabilities of existing models to construct optimal condition pairs, which are fed into VACE (Jiang et al., 2025) for edited video generation. For each editing task, specific strategy is used for condition generation.

**Object Removal.** Inputting the masked video (derived from object masks) and target prompt into VACE for edited video generation often fails to eliminate the object cleanly or leads to the hallucination of unexpected contents. To mitigate this issue, we adopt a two-stage approach for edited video generation. First, we employ ObjectClear (Zhao et al., 2026) on the first frame to perform clean object removal. Next, we concatenate this edited frame with the subsequent masked video frames, which is then fed into VACE to perform video inpainting, yielding stable and high-quality object removal.

**Object Addition.** We treat this task as the inverse of object removal. Once a valid removal pair is generated, we simply swap the source and edited videos to create a corresponding training pair for the object addition task.

**Object Replacement.** VACE demonstrates robust performance on object replacement. Therefore, we simply feed the masked video sequence and the target video prompt into VACE to generate high-quality replacement results.

**Video Stylization.** Although VACE supports video stylization conditioned on depth maps, maintaining the content structure of the original video is not satisfactory. Thus, we employ a strategy similar to object removal. We first utilize FLUX.1 Kontext (Labs et al., 2025) to apply the style transfer on the first frame. Subsequently, we concatenate the edited frame with the depth map sequence (extracted via MiDaS (Ranftl et al., 2020)) to serve as the input condition pair for VACE, thereby generating a temporally consistent stylized video. Specifically designed for video stylization, our pipeline addresses common artifacts in VACE-generated data, such as frame collapse, abrupt transitions, temporal inconsistency, and content distortion (e.g., facial deformations). To ensure high-quality output, we implement a two-stage strategy. First, we use Qwen-3-VL-Instant (Bai et al., 2025b) to filter for smooth and stable videos, removing low-quality frames with severe artifacts or flickering. Second, we refine the selected videos using the 14B Wan-2.2-T2V (Wan et al., 2025) model. These combined strategies enable the synthesis of stylized videos with significantly improved visual and temporal quality.

## A.5. Video Synthesis

Once the requisite condition pairs are prepared, we execute VACE in large-scale batches to synthesize high-quality editing videos. To maximize the utility of the synthesized data and ensure efficient construction, we design a data augmentation strategy to generate additional training pairs without extra computational cost.

**Reversible Replacement.** For the object replacement based on one source video, we treat such process as reversible. By swapping the source and target videos, we effectively double the volume of the replacement data.

**Cross-Task Augmentation.** In fact, the edited videos generated from object removal and replacement share the same clean background. Therefore, the synthesized video from the replacement (containing a novel object) can be paired with the background video from the removal task. This allows us to construct new "removal" pairs (new object → background) and "adding" pairs (background → new object), effectively doubling the dataset size for both tasks.

Finally, we totally construct approximately 800K video pairs for the four editing tasks. The entire data synthesis process required approximately **76,800 GPU hours** on NVIDIA RTX 4090.

## A.6. Video Filtering and Re-captioning

To pursue high quality of instruction-video pairs, we employ the VLLM, i.e., Gemini-2.5-Flash-Thinking, to evaluate and filter out low-quality samples in total 800K video pairs. We extract representative key frames from the source and edited

videos, and concatenate them into a side-by-side layout to facilitate VLLM assessment. The remained video pairs are re-captioned by VLLM. The entire caption process (including Sec. A.3) incurred a total cost of approximately $13,600. Finally, we construct ReCo-Data with 500K high-quality instruction-video pairs. Each video clip contains 81 frames with the resolution of $480 \times 832$ and duration of 5 seconds.

## B. VLLM-based Evaluation Benchmark

Traditional video generation metrics often struggle to accurately assess the fidelity and quality of video editing. Inspired by recent image editing evaluation protocols (Wei et al., 2025), we propose a VLLM-based evaluation benchmark to comprehensively and effectively assess video editing quality.

**Testing Data.** We collect 480 video-instruction pairs as the testing data, distributed evenly with 120 pairs for each of the four tasks (i.e., object add, remove, replace, and video stylization). All source videos are collected from Pexels video platform. For local editing tasks (i.e., object add, remove and replace), we utilize Gemini-2.5-Flash-Thinking (Anil et al., 2023) to brainstorm and generate diverse editing instructions based on the video content. For rigorous evaluation on video stylization, we randomly select 10 source videos and apply 12 distinct styles to each, resulting in 120 evaluation pairs.

**Evaluation Metrics.** While previous image-based metrics primarily focus on editing accuracy and static generation quality, evaluating video editing entails greater complexity. To address this, we construct a diverse set of evaluation dimensions specifically tailored for video. Corresponding system prompt designed for the VLLM is presented in Figure 9, which evaluates performance across three major perspectives, comprising a total of nine sub-dimensions:

- **Edit Accuracy** ($S_{EA}$): evaluate how well the result aligns with the instruction.
    - Semantic Accuracy (SA): Does the edited video correctly follow the semantics of the text instruction?
    - Scope Precision (SP): Is the editing confined strictly to the target region without affecting the background?
    - Content Preservation (CP): Are the non-edited regions or original details faithfully preserved? (For stylization, this corresponds to structural preservation.)

- **Video Naturalness** ($S_{VN}$): evaluates the realism and coherence of the generated content.
    - Appearance Naturalness (AN): Are the lighting, texture, and color of the edited video natural?
    - Scale Naturalness (SN): Is the size and proportion of the edited object reasonable relative to the environment? (For stylization, this captures cases where the stylized object becomes unreasonably large.)
    - Motion Naturalness (MN): Does the movement of the edited object (or the style rendering) follow physically plausible dynamics?

- **Video Quality** ($S_{VQ}$): evaluates the fundamental visual quality of the edited video.
    - Visual Fidelity (VF): Is the video clear, sharp, and free from visual artifacts?
    - Temporal Stability (TS): Is the video free from flickering or jittering across frames?
    - Edit Stability (ES): Is the edited content consistently preserved in identity and appearance throughout the video duration?

The VLLM rates the score for each sub-dimension from 0 to 10. Then, we attain the per-category scores (i.e., $S_{EA}$, $S_{VN}$, $S_{VQ}$) by calculating the geometric mean of their respective sub-dimensions as:

$$S_{EA} = \sqrt[3]{SA \cdot SP \cdot CP}, \tag{18}$$

$$S_{VN} = \sqrt[3]{AN \cdot SN \cdot MN}, \tag{19}$$

$$S_{VQ} = \sqrt[3]{VF \cdot TS \cdot ES}. \tag{20}$$

Finally, the overall score $S$ is calculated as the arithmetic mean of the three per-category scores:

$$S = \frac{1}{3}\big(S_{EA} + S_{VN} + S_{VQ}\big). \tag{21}$$

**Human:**
You are a professional digital artist and video quality evaluator. Your task is to evaluate an AI-generated video edit based on three major categories: Edit Accuracy, Video Quality, and Naturalness. You will be given the text instruction used to create the edit and side-by-side video keyframes, where the left side shows the original video and the right side shows the edited version. You must provide your output only in the following JSON format. Do not output anything else.
{
  "edit_accuracy": {"scores": [1, 1, 1], "reasoning": "..."},
  "video_quality": {"scores": [1, 1, 1], "reasoning": "..."},
  "naturalness": {"scores": [1, 1, 1], "reasoning": "..."}
}
Keep each reasoning string concise and short, summarizing the scores for that category.

**Category 1: Edit Accuracy**
This category evaluates how well the AI understood and executed the text instruction. The scores list for edit_accuracy contains three scores:
**[Score_SA, Score_SP, Score_CP]**.

- Score SA: Semantic Accuracy (Scale: 1-10). Rates if the core concept of the edit is correct (e.g., what was added, removed, replaced, or stylized). 1 means the core concept is completely wrong. 10 means the core concept perfectly matches the instruction.
- Score SP: Scope Precision (Scale: 1-10). Rates if the location, area, or scope of the edit is correct (e.g., where the edit was applied). 1 means the location/area is completely wrong. 10 means the edit is perfectly localized or globalized exactly as instructed.
- Score CP: Content Preservation (Scale: 1-10). Rates if the AI negatively affected areas that should not have been edited. 1 means unedited areas are heavily distorted or changed. 10 means all content outside the specified edit scope is perfectly preserved. For global stylization, this evaluates if the underlying structure is preserved.

**Category 2: Video Quality**
This category evaluates the technical fidelity and stability of the edited video. The scores list for video_quality contains three scores:
**[Score_VF, Score_TS, Score_ES]**.

- Score VF: Visual Fidelity (Scale: 1-10). Rates the overall clarity and presence of static visual artifacts in the edited frames. 1 means the video is extremely blurry or full of artifacts. 10 means the video is sharp and clear.
- Score TS: Temporal Stability (Pixel-level) (Scale: 1-10). Rates the low-level consistency of the video over time, focusing on flicker or boiling textures. 1 means the video is extremely unstable. 10 means the video is perfectly stable over time.
- Score ES: Edit Effect Persistence (Semantic-level) (Scale: 1-10). Rates if the intended edit effect (add, remove, replace, style) is stable and persists correctly from beginning to end. 1 means the edit effect fails mid-video. 10 means the intended edit effect is perfectly stable. Bad cases include removed objects popping back or added objects disappearing.

**Category 3: Visual Naturalness**
This category evaluates how plausible and seamlessly integrated the edit is. The scores list for naturalness contains three scores:
**[Score_AN, Score_SN, Score_MN]**.

- Score AN: Appearance Naturalness (Integration) (Scale: 1-10). Rates how naturally the new parts blend with the original scene's lighting and shadows. 1 means the edit looks fake and pasted on. 10 means the edit is perfectly integrated.
- Score SN: Scale & Proportion (Scale: 1-10). Rates if the edited object's size is reasonable. 1 means the scale is illogical. 10 means the size is perfectly proportional. Bad cases include giant pets or magnified background textures after removal.
- Score MN: Motion Naturalness (Physical Laws) (Scale: 1-10). Rates if the edit obeys basic physics and interacts logically. 1 means the edit violates physics. 10 means the behavior is physically plausible. Bad cases include objects defying gravity or not tracking movement correctly.

**Critical Rule: Failed Edits (Identical Videos)**
If the edited video (right side) is identical to the original video (left side), this indicates a total failure. You must set all nine scores to 0.
Example output for failure:
{
  "edit_accuracy": { "scores": [0, 0, 0], "reasoning": "Edit failed to apply. Identical videos." },
  "video_quality": { "scores": [0, 0, 0], "reasoning": "Edit failed to apply. Identical videos." },
  "naturalness": { "scores": [0, 0, 0], "reasoning": "Edit failed to apply. Identical videos." }
}

Editing instruction: <EDITING INSTRUCTION>
<Video> Source Video </Video>
<Video> Edited Video </Video>

**Assistant:**

*Figure 9.* The system prompts that are fed into Gemini-2.5-Flash-Thinking for video editing assessment. We require VLLM to evaluate the four video editing tasks from three major perspectives, i.e., edit accuracy, video naturalness and video quality.

*Table 4.* Performance comparisons across different video editing models under traditional evaluation metrics without using VLLMs.

| Task | Approach | CLIP-T (Frame)↑ | CLIP-T (Video)↑ | Motion Smoothness↑ | Aesthetic Quality↑ | Dynamic Degree↑ |
|---|---|---|---|---|---|---|
| Add | InsViE | 24.98 | 20.72 | 97.969 | 0.523 | 0.360 |
| | Lucy-Edit | 25.87 | 20.69 | 99.060 | 0.559 | 0.380 |
| | Ditto | 25.78 | 20.12 | **99.145** | **0.591** | **0.410** |
| | ReCo | **25.94** | **21.22** | 99.067 | 0.582 | 0.390 |
| Replace | InsViE | 21.81 | 17.73 | 97.728 | 0.526 | 0.317 |
| | Lucy-Edit | 24.84 | 19.86 | 99.073 | 0.358 | 0.358 |
| | Ditto | 24.29 | 19.41 | **99.161** | **0.600** | **0.392** |
| | ReCo | **25.55** | **21.38** | 99.144 | 0.553 | 0.275 |
| Remove | InsViE | 23.03 | 18.36 | 97.810 | 0.469 | 0.308 |
| | VACE | 24.27 | 19.27 | 99.111 | **0.556** | **0.342** |
| | ReCo | **24.68** | **20.78** | **99.299** | 0.483 | 0.233 |
| Style | InsViE | 24.79 | 17.74 | 97.567 | 0.549 | 0.390 |
| | Lucy-Edit | 22.00 | 15.33 | **99.031** | 0.563 | 0.380 |
| | Ditto | 27.43 | 19.52 | 98.161 | **0.620** | **0.410** |
| | ReCo | **27.57** | **19.98** | 97.955 | 0.616 | 0.370 |

*Table 5.* Human evaluation of pick ratio over four instructional video editing tasks. "-" indicates that the model lacks the capability for that specific editing task.

| Approach | Add | Replace | Remove | Style |
|---|---|---|---|---|
| InsViE | 2.5% | 5.0% | 2.5% | 5.0% |
| VACE | – | – | 32.5% | – |
| Lucy-Edit | 25.0% | 22.5% | – | 2.5% |
| Ditto | 12.5% | 10.0% | – | 40.0% |
| ReCo | **60.0%** | **62.5%** | **65.0%** | **52.5%** |

## C. Implementation Details of Baselines and ReCo

**Baseline Settings.** For recent video editing advances, few methods possess the versatility to handle all four editing tasks simultaneously. Here, we outline the criteria for our baseline selection. For **object addition, replacement, and video stylization**, we benchmark against InsViE (Wu et al., 2025b), Lucy-Edit (DecartAI Team, 2025), and Ditto (Bai et al., 2025a). Since InsViE is constrained to an input of 49 frames at $480 \times 720$ resolution, we adapt our test videos via uniform temporal down-sampling and spatial resizing to match the requirements. For the **object removal**, effective instruction-based baselines are scarce. To facilitate a meaningful comparison, we include VACE (Jiang et al., 2025) as an additional baseline. Unlike instruction-based methods, VACE requires both an explicit object mask and a target video prompt to perform removal. Note that VACE exhibits some instability in this implementation.

**Implementation Details of ReCo.** ReCo is built upon the Wan (Wan et al., 2025) architecture and trained on ReCo-Data using the AdamW optimizer. We employ a two-stage learning rate schedule: an initial phase with a learning rate of $1 \times 10^{-4}$ to ensure stable convergence, followed by a fine-tuning phase at $2 \times 10^{-5}$ for precise refinement. Regarding the loss weights, the initial values of the latent and attention constraints typically fall within $[-1, 1]$, whereas the MSE loss of flow matching is approximately $0.03$. To balance the impacts of gradients, we scale the magnitude of each region-constraint loss to be roughly $0.1\times$ that of the MSE loss. Consequently, we set the weighting coefficients ($\lambda_1$ and $\lambda_2$) as $1 \times 10^{-3}$. All experiments were conducted on a cluster of $24$ NVIDIA A800 GPUs with a total mini-batch size of $24$, requiring about $10$ days for training.

## D. More Evaluations on ReCo

### D.1. Traditional Evaluation Metrics

Table 4 lists the performance comparisons under traditional evaluation metrics which are not based on VLLMs. We follow the protocols in recent video editing advance (Bai et al., 2025a; Zi et al., 2025; Jiang et al., 2025) and VBench (Huang

**Gemini-2.5-Flash-Thinking**

| Task | Approach | $S_{EA}$ | $S_{VN}$ | $S_{VQ}$ | $S$ |
|---|---|---|---|---|---|
| Add | InsViE | 2.60 | 3.10 | 3.46 | 3.05 |
| | Lucy-Edit | 6.47 | 5.70 | 6.77 | 6.31 |
| | Ditto | 6.70 | **7.57** | 8.41 | 7.56 |
| | ReCo | **8.54** | 7.55 | **8.61** | **8.23** |
| Replace | InsViE | 2.10 | 3.91 | 3.49 | 3.17 |
| | Lucy-Edit | 7.08 | 6.21 | 6.88 | 6.72 |
| | Ditto | 4.56 | 7.21 | 7.96 | 6.58 |
| | ReCo | **9.43** | **8.01** | **8.77** | **8.74** |
| Remove | InsViE | 2.44 | 3.76 | 3.29 | 3.16 |
| | VACE | 4.57 | 5.43 | 5.56 | 5.19 |
| | ReCo | **7.28** | **6.90** | **6.82** | **7.00** |
| Style | InsViE | 8.17 | 8.21 | 7.35 | 7.91 |
| | Lucy-Edit | 4.65 | 4.67 | 5.17 | 4.83 |
| | Ditto | 9.20 | 9.07 | 8.77 | 9.01 |
| | ReCo | **9.42** | **9.19** | **8.90** | **9.17** |

**GPT-4o**

| Task | Approach | $S_{EA}$ | $S_{VN}$ | $S_{VQ}$ | $S$ |
|---|---|---|---|---|---|
| Add | InsViE | 1.47 | 1.47 | 1.54 | 1.49 |
| | Lucy-Edit | 4.71 | 4.59 | 4.74 | 4.68 |
| | Ditto | 6.86 | 6.71 | 6.83 | 6.80 |
| | ReCo | **7.50** | **7.35** | **7.64** | **7.50** |
| Replace | InsViE | 1.07 | 1.31 | 1.35 | 1.24 |
| | Lucy-Edit | 5.55 | 5.43 | 5.59 | 5.52 |
| | Ditto | 3.67 | 3.84 | 3.92 | 3.81 |
| | ReCo | **7.51** | **7.16** | **7.36** | **7.34** |
| Remove | InsViE | 0.27 | 0.28 | 0.27 | 0.28 |
| | VACE | 0.68 | 0.66 | 0.66 | 0.67 |
| | ReCo | **1.00** | **0.97** | **0.94** | **0.97** |
| Style | InsViE | 5.26 | 5.00 | 4.81 | 5.02 |
| | Lucy-Edit | 0.02 | 0.02 | 0.02 | 0.02 |
| | Ditto | 7.69 | 7.46 | 7.39 | 7.52 |
| | ReCo | **8.32** | **8.03** | **7.99** | **8.11** |

**Kimi-k2.5**

| Task | Approach | $S_{EA}$ | $S_{VN}$ | $S_{VQ}$ | $S$ |
|---|---|---|---|---|---|
| Add | InsViE | 2.08 | 2.23 | 2.44 | 2.25 |
| | Lucy-Edit | 5.94 | 5.14 | 5.80 | 5.63 |
| | Ditto | 7.72 | 6.84 | 7.51 | 7.36 |
| | ReCo | **8.70** | **7.36** | **8.32** | **8.13** |
| Replace | InsViE | 1.90 | 2.34 | 2.60 | 2.28 |
| | Lucy-Edit | 6.58 | 5.54 | 6.40 | 6.17 |
| | Ditto | 6.09 | 5.96 | 6.95 | 6.33 |
| | ReCo | **9.37** | **7.34** | **8.51** | **8.41** |
| Remove | InsViE | 0.96 | 1.47 | 1.38 | 1.27 |
| | VACE | 4.55 | 4.60 | 4.81 | 4.65 |
| | ReCo | **8.39** | **7.86** | **7.79** | **8.01** |
| Style | InsViE | 8.18 | 7.05 | 7.10 | 7.44 |
| | Lucy-Edit | 2.22 | 2.82 | 2.45 | 2.50 |
| | Ditto | 8.95 | 7.87 | **8.14** | 8.32 |
| | ReCo | **9.01** | **7.89** | 8.09 | **8.33** |

**Qwen-3.5-Plus**

| Task | Approach | $S_{EA}$ | $S_{VN}$ | $S_{VQ}$ | $S$ |
|---|---|---|---|---|---|
| Add | InsViE | 1.89 | 2.63 | 3.28 | 2.60 |
| | Lucy-Edit | 4.98 | 4.52 | 5.71 | 5.07 |
| | Ditto | 4.41 | 6.25 | 7.60 | 6.09 |
| | ReCo | **7.47** | **6.60** | **8.43** | **7.50** |
| Replace | InsViE | 2.01 | 3.18 | 2.92 | 2.70 |
| | Lucy-Edit | 5.46 | 5.10 | 6.46 | 5.67 |
| | Ditto | 2.87 | 6.31 | 7.64 | 5.61 |
| | ReCo | **8.63** | **6.72** | **8.65** | **8.00** |
| Remove | InsViE | 0.89 | 3.44 | 2.10 | 2.14 |
| | VACE | 4.14 | 6.70 | 6.24 | 5.69 |
| | ReCo | **6.80** | **6.81** | **7.39** | **7.00** |
| Style | InsViE | 7.04 | 7.11 | 6.88 | 7.01 |
| | Lucy-Edit | 0.96 | 1.99 | 1.39 | 1.44 |
| | Ditto | 9.12 | 9.21 | 9.08 | 9.14 |
| | ReCo | **9.22** | **9.38** | **9.22** | **9.27** |

*Figure 10.* Evaluation bias analysis with different VLM evaluators. The consistent trends across Gemini-2.5-Flash-Thinking, GPT-4o, Kimi-K2.5, and Qwen-3.5-Plus suggest that the potential evaluator-specific bias is limited.

et al., 2024b), and choose the CLIP-T scores at frame and video level, motion smoothness, aesthetic quality and dynamic degree as the measurements. Overall, our ReCo attains the best text-video alignment (i.e., CLIP-T) across all editing tasks, demonstrating a robust ability to follow instructions. Besides, ReCo delivers competitive temporal coherence, characterized by stable motion smoothness and significantly reduced temporal flickering, while simultaneously attaining high aesthetic quality. Although Ditto occasionally achieves higher aesthetic scores, the qualitative analysis in Figure 4 reveals a tendency to over-stylize video content during local editing, often at the expense of the original video's integrity. Our ReCo, instead, executes the intended edits faithfully, balancing competitive aesthetics with better temporal stability and visual fidelity.

### D.2. Human Evaluation

To evaluate the user preference across different video editing models, we further conducted a human evaluation. We randomly select 40 videos per task from the test set, and invite 10 volunteers for evaluation. In each trial, evaluators are provided with the source video and the corresponding editing instruction, followed by the edited videos by all compared models. To ensure a fair comparison, the model identities are anonymized, and the display order is randomized. All evaluators are asked to select the best video based on editing fidelity and overall quality.

As shown in Table 5, the edited videos by ReCo are picked by the majority of users. Meanwhile, the statistical trends of our VLLM-based evaluations closely align with the human study results. This high correlation validates the rationality of our proposed VLLM-based metrics and benchmark, manifesting the strong alignment with human preferences.

*Table 6.* Robustness to training mask padding. We report the overall score $S$ under different random padding ranges for the edited-region masks. ReCo achieves similar performance with 5–10 and 10–15 pixel padding, while overly large padding, e.g., 55–60 pixels, degrades performance by including excessive background regions into the editable area.

| Training Mask Padding | Add | Replace | Remove | Style |
|---|---|---|---|---|
| ReCo (5–10 px) | **8.23** | **8.74** | 7.00 | **9.17** |
| ReCo (10–15 px) | 8.19 | 8.63 | **7.04** | 9.13 |
| ReCo (55–60 px) | 7.96 | 8.12 | 6.63 | 9.08 |

**Edited Video Pair**  **Edited Region Mask**

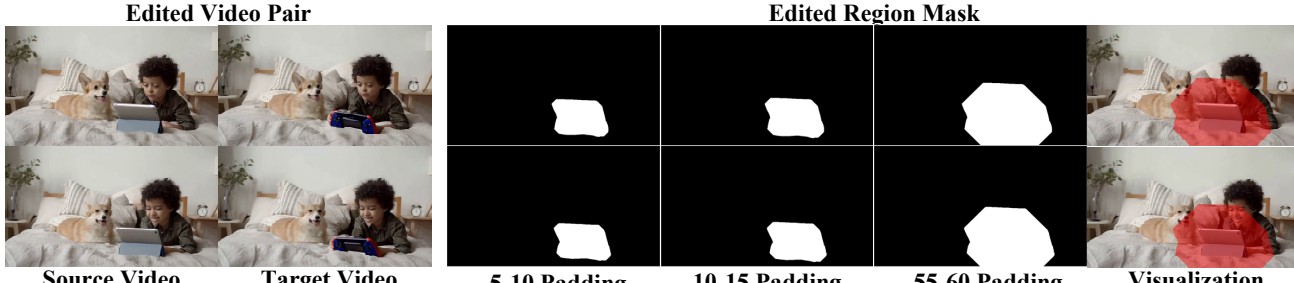

Source Video   Target Video      5-10 Padding   10-15 Padding   55-60 Padding   Visualization

*Figure 11.* Illustration of the training video pair (source video and edited target video) and the corresponding edited-region masks under different padding settings, including 5–10, 10–15, and 55–60 pixels. We particularly highlight the mask overlay in the 55–60 padding case. As the padding becomes too large, the mask covers not only the intended editing region but also important background content outside the edit area.

### D.3. VLM Evaluator Bias Analysis

Since our benchmark uses Gemini-2.5-Flash as the primary VLM evaluator, there may be concerns about potential model-specific bias, especially because Gemini is also involved in data filtering. To examine this issue, we further evaluate ReCo with four additional widely used VLMs, including Gemini-2.5-Flash-Thinking, GPT-4o, Kimi-K2.5, and Qwen-3.5-Plus. As shown in Figure 10, the evaluation results exhibit highly consistent trends across different VLM evaluators, suggesting that the potential evaluator-specific bias is limited. Moreover, these trends are also consistent with our 10-person human evaluation reported in the supplementary material, further supporting the reliability and validity of the proposed benchmark.

### D.4. Robustness to Training Masks

In general, the edited-region mask used in our regional constraint during training covers the entire edited region, i.e., the union of the source object and the target object. To mitigate noisy boundaries introduced by segmentation models (e.g., SAM2), we apply random padding of 5–10 pixels by default, following the common practice in existing inpainting settings. Consistent with data construction, we use the same edited-area masks with 5–10 pixel padding by default during training, covering both the original object region and the target region.

To further study the sensitivity to mask size and mask accuracy, we evaluate several mask variants, as illustrated in Figure 11. As shown in Table 6, the performance remains very similar when the padding is 5–10 pixels or 10–15 pixels. It supports our hypothesis that ReCo training does not require highly accurate masks. Instead, the regional constraint mainly helps the model learn an implicit localization of the editing region. However, when the padding becomes too large (e.g., 55–60 pixels), more background objects that should otherwise be preserved are incorrectly included in the edit area. This issue is particularly harmful for small-object editing, as shown in Figure 12. When trained with such regional constraints, the model tends to modify content outside the intended editing region, and inaccurate localization may undesirably affect surrounding areas.

### D.5. Memory and Latency Overhead of In-Context Generation

We benchmark the inference cost of editing videos with resolution 480×832×81 on a single A100 GPU. Table 7 compares the latency and peak memory among ReCo, several baselines, and Wan+ControlNet discussed in the main paper. Lucy-Edit and InsViE are faster, but their architectures are not well suited for unifying diverse editing tasks, which leads to poor performances. Similarly, Ditto and Wan+ControlNet, directly exploiting reference video as ControlNet condition, also

*Table 7.* Inference cost comparison on video editing at resolution $480 \times 832 \times 81$ using a single A100 GPU. We report task performance, model parameters, latency, peak memory, and FLOPs. ReCo achieves the best overall editing performance with a compact 2.1B-parameter model and substantially lower memory usage than Ditto.

| Model | Add | Replace | Remove | Style | Params | Latency (s) | Peak Mem (GB) | FLOPs |
|---|---|---|---|---|---|---|---|---|
| InsViE | 3.05 | 3.17 | 3.16 | 7.91 | 6.9B | 126 | 35.4 | 18.3P |
| Lucy-Edit | 6.31 | 6.72 | – | 4.83 | 5B | **86** | 29.2 | **10.6P** |
| Ditto | 7.56 | 6.58 | – | 9.01 | 17.3B | 1,023 | 64.3 | 95.9P |
| Wan+ControlNet | 7.43 | 8.07 | 6.58 | 8.73 | **2.1B** | 280 | **22.8** | 14.7P |
| ReCo | **8.23** | **8.74** | **7.00** | **9.17** | **2.1B** | 533 | 26.9 | 90.6P |

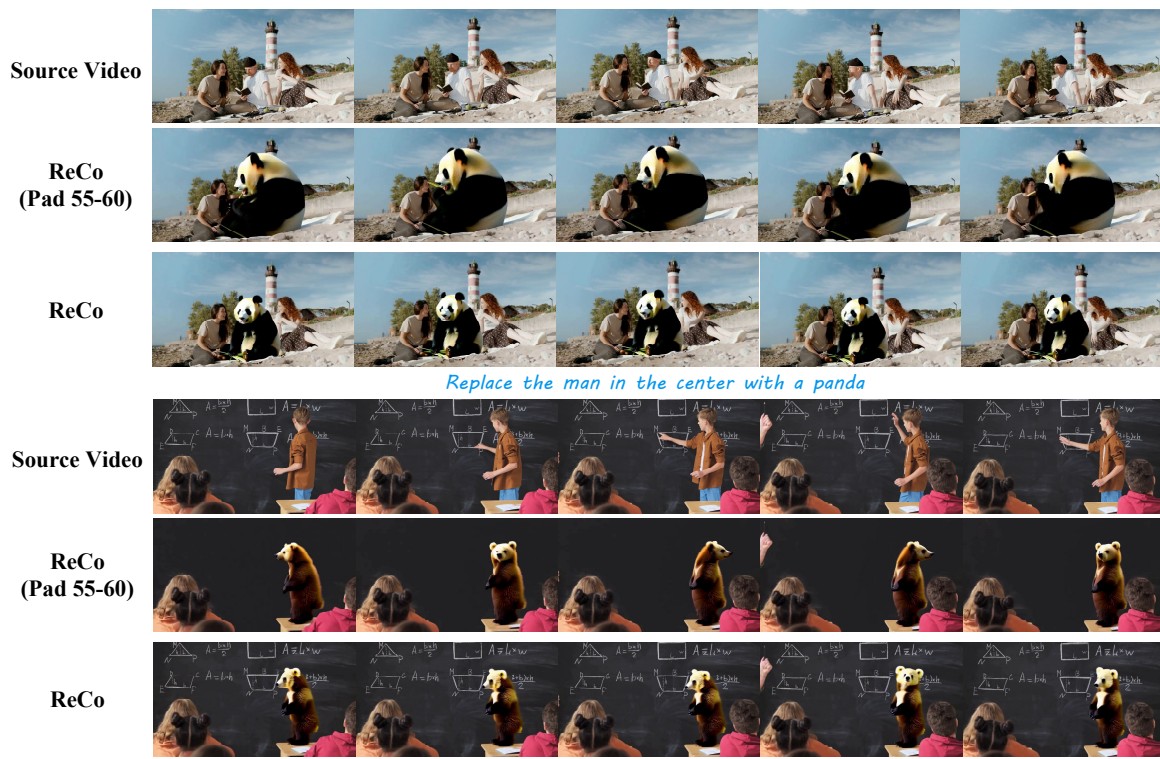

*Figure 12.* Comparison between ReCo trained with large-mask regional constraints (55–60 pixel padding) and the standard ReCo setting (5–10 pixel padding).

struggle to support multi-task editing within one model. In contrast, ReCo reformulates video editing as an in-context generation paradigm, providing a more unified framework for multi-task video editing and yielding better performance. ReCo has only 2.1B parameters, making it one of the smallest models in the comparison and about 8.2× smaller than Ditto (17.3B). It requires only 26.9 GB of inference memory, compared with 64.3 GB for Ditto, making it much more friendly to consumer GPUs. Although ReCo has somewhat higher inference latency, it achieves the best overall trade-off between model size, memory usage, and performance. Regarding training efficiency, ReCo converges in about 6 days on 24 A100 GPUs, using 66.2 GB training memory at a global batch size of 24. By comparison, Wan+ControlNet requires 48 GB of training memory.

### D.6. Implicit Region Grounding during Inference

ReCo only uses masks during training and requires no explicit mask in the inference stage. Specifically, the regional constraint loss in our ReCo encourages the model to allocate greater attention to the regions that should be edited. As a result, during inference, ReCo can directly identify the target editing regions and their associated attention regions for content modification. We view this as evidence that ReCo has developed an implicit grounding capability. The results are demonstrated in Figure 13.

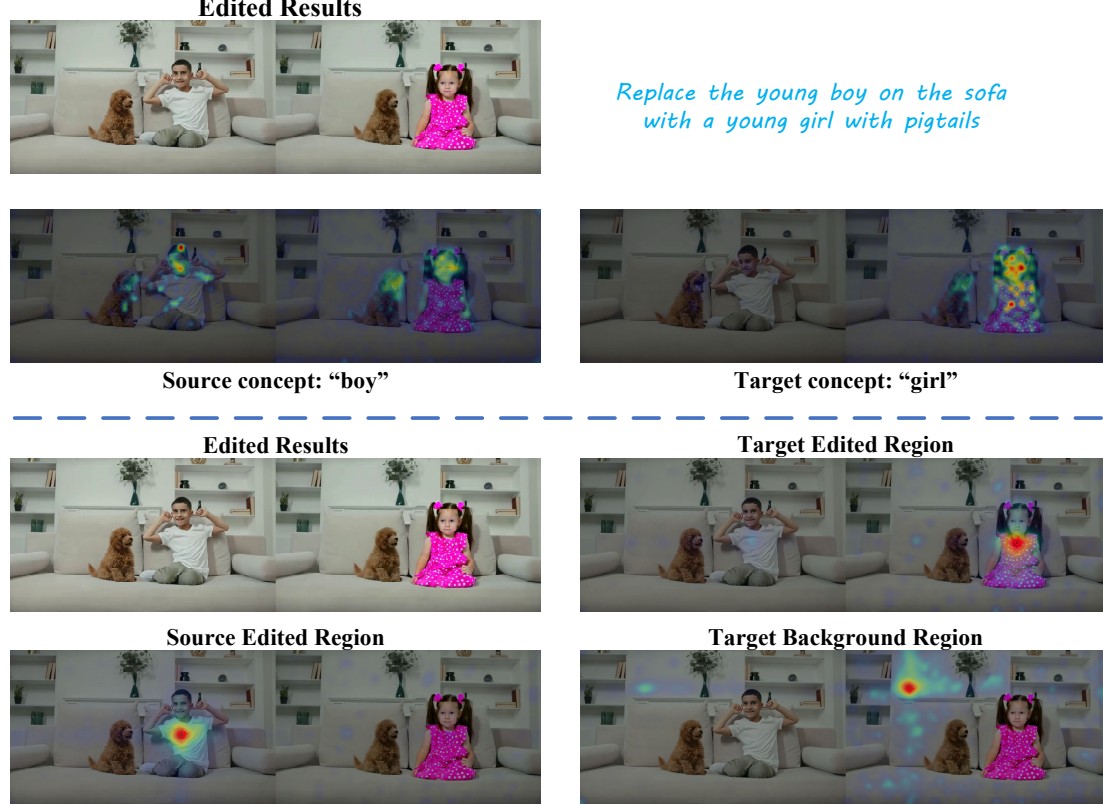

*Figure 13.* Visualization of the cross-attention and self-attention maps of ReCo during inference. Top: cross-attention maps showing how the full video latent attends to the source and target prompts. Bottom: self-attention maps showing how sampled points from the target edited region, source edited region, and target background region attend to the full video latent.

## E. Additional Analysis and Qualitative Results of ReCo

### E.1. Generalization Ability of ReCo

Interestingly, Figure 14 shows that ReCo can generalize to abstract and creative editing tasks. For instance, it successfully synthesizes a halo on a woman's head, generates a cascading confetti effect, places an "idea lightbulb" beside a man's head, and creates smoke emitting from a computer. We attribute such generalization ability of ReCo to effectively inheriting and leveraging the rich priors from the pre-trained video diffusion model.

### E.2. Complex Editing Scenarios

We provide two more challenging scenarios, i.e., video editing with large motion and multiple objects in Figure 15. Even in such challenging settings, our ReCo can still perform stable video editing.

### E.3. Extension to Longer Video Editing

The key challenge in long-video editing is maintaining temporal appearance and motion coherence across clips within the edited regions. Since our ReCo is fine-tuned based on VACE, it can naturally support temporal extension for long video editing. Specifically, we split the long video into overlapping clips (e.g., 5s, 4s, and 4s). The first clip is edited normally. For each subsequent clip, we employ the last 16 edited frames from the previous clip as condition, where the in-context input is formed as [source video, (16 edited frames + 65 masked video frames)]. Repeating this process across all subsequent clips enables editing of long videos.

As shown in Figure 16, after a simple fine-tuning stage, our model can successfully perform replacement editing on a 10-second video while preserving good continuity across clip boundaries.

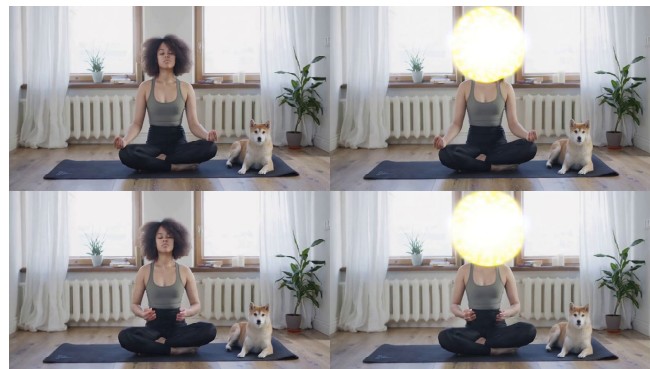

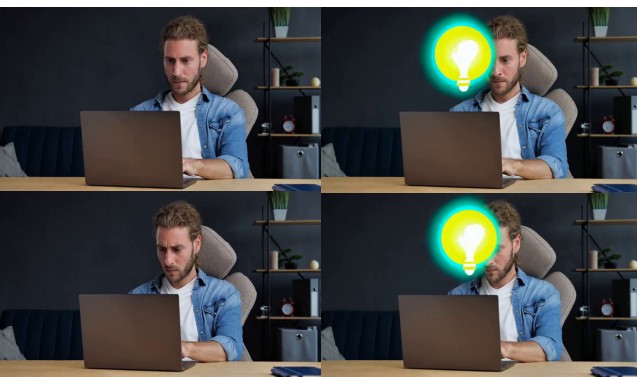

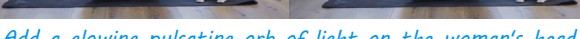

*Add a glowing pulsating orb of light on the woman's head*

*Add a bright lightbulb icon appearing beside the man's head*

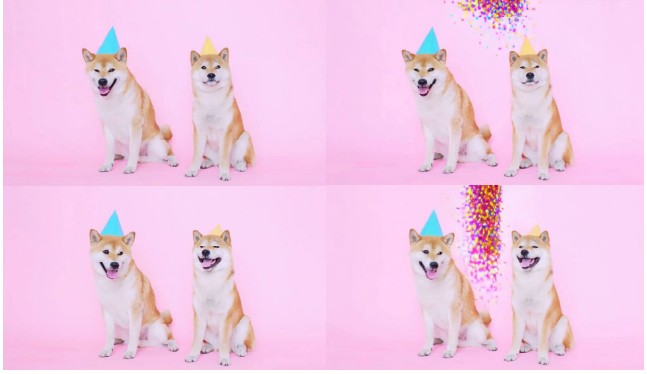

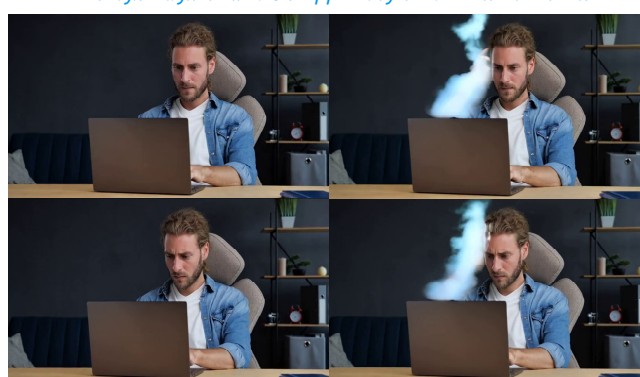

*Add colorful confetti falling from the top of the screen*

*Add smoke rising from the computer*

*Figure 14.* Four examples of instructional video editing by ReCo to verify the generalization ability. Our model demonstrates the strong generalization to the abstract and creative editing tasks.

### E.4. Shadows and Reflections in Video Editing

Reflections, relighting, and shadow-related effects are widely recognized as particular challenges in image/video editing, and it is difficult to fully address them. We also make every effort to account for these factors in data construction.

As illustrated in Figure 17, for object removal, we first edit the first frame with ObjectClear, which can simultaneously remove the object and its associated shadows. We then compute a diff mask against the original frame and use it to guide SAM2 tracking across the video. The resulting edited-region mask sequence is further used by VACE to remove the object together with its shadows and reflections. Since the mask covers both the object and its associated effects, these regions are treated as part of the edited area rather than the preserved background, and our latent-space regularization therefore does not suppress dynamic shadow or reflection generation for newly edited objects. For the add task, we simply reverse the original and edited videos during training while keeping the same edited-region mask, which similarly allows the model to learn such effects.

For object replacement, we don't use the same pipeline because reliable shadow-/reflection-aware mask-based image replacement models are still lacking, and existing instruction-based models are often inaccurate enough for stable data construction. As a compromise, we use the original mask with larger padding to cover as much of the shadow/reflection region as possible.

With the resulting add, remove, and replace training data, the model can handle some general cases, as shown in Figure 18. However, clear limitations remain for more complex scenarios, especially when shadows are long-range or far from the object, as shown in Figure 19.

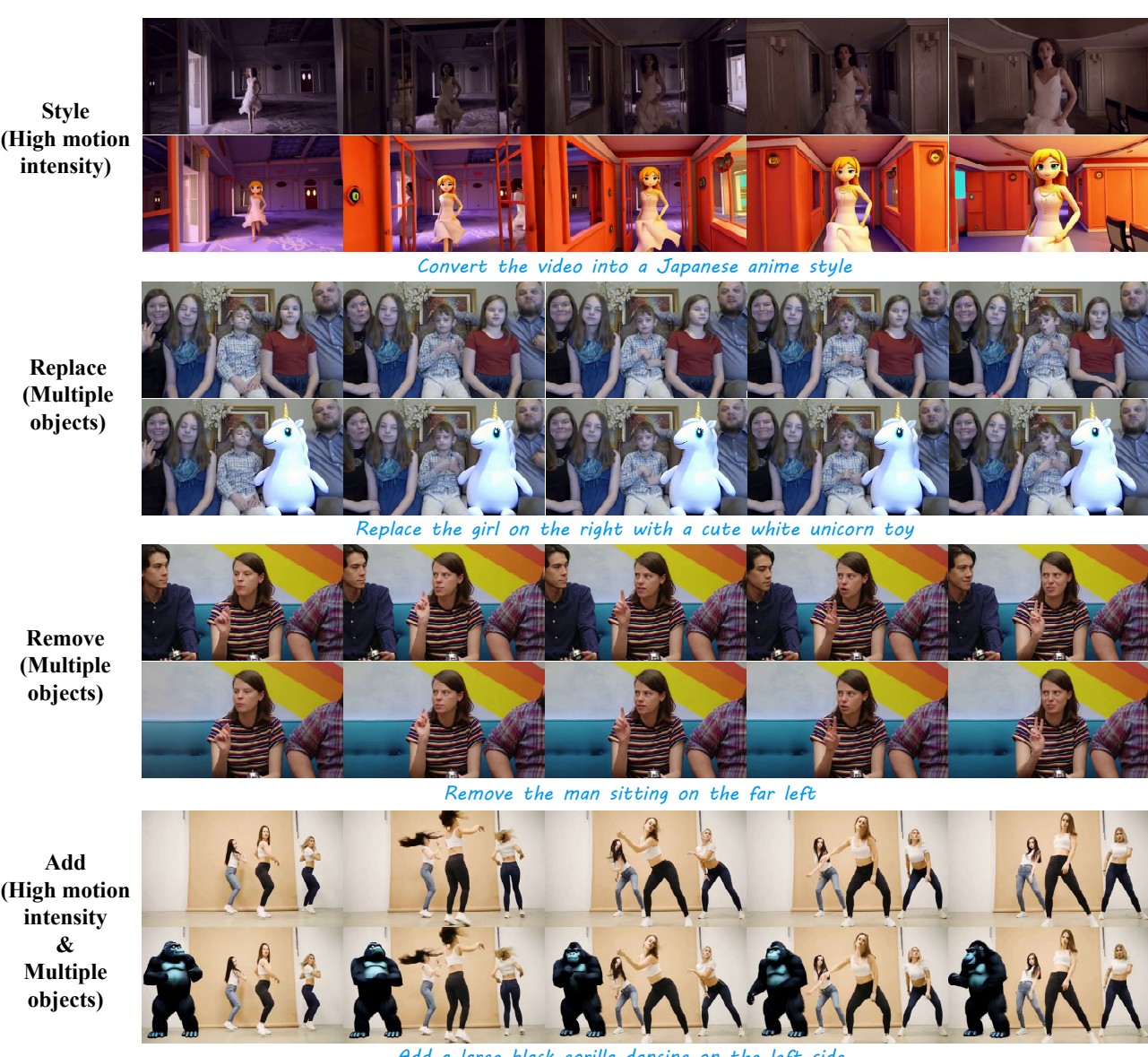

*Figure 15.* Examples of video editing results for four tasks in complex scenes, with a focus on highly dynamic, multi-object scenarios. For each example, the top row shows the original video and the bottom row shows the edited video.

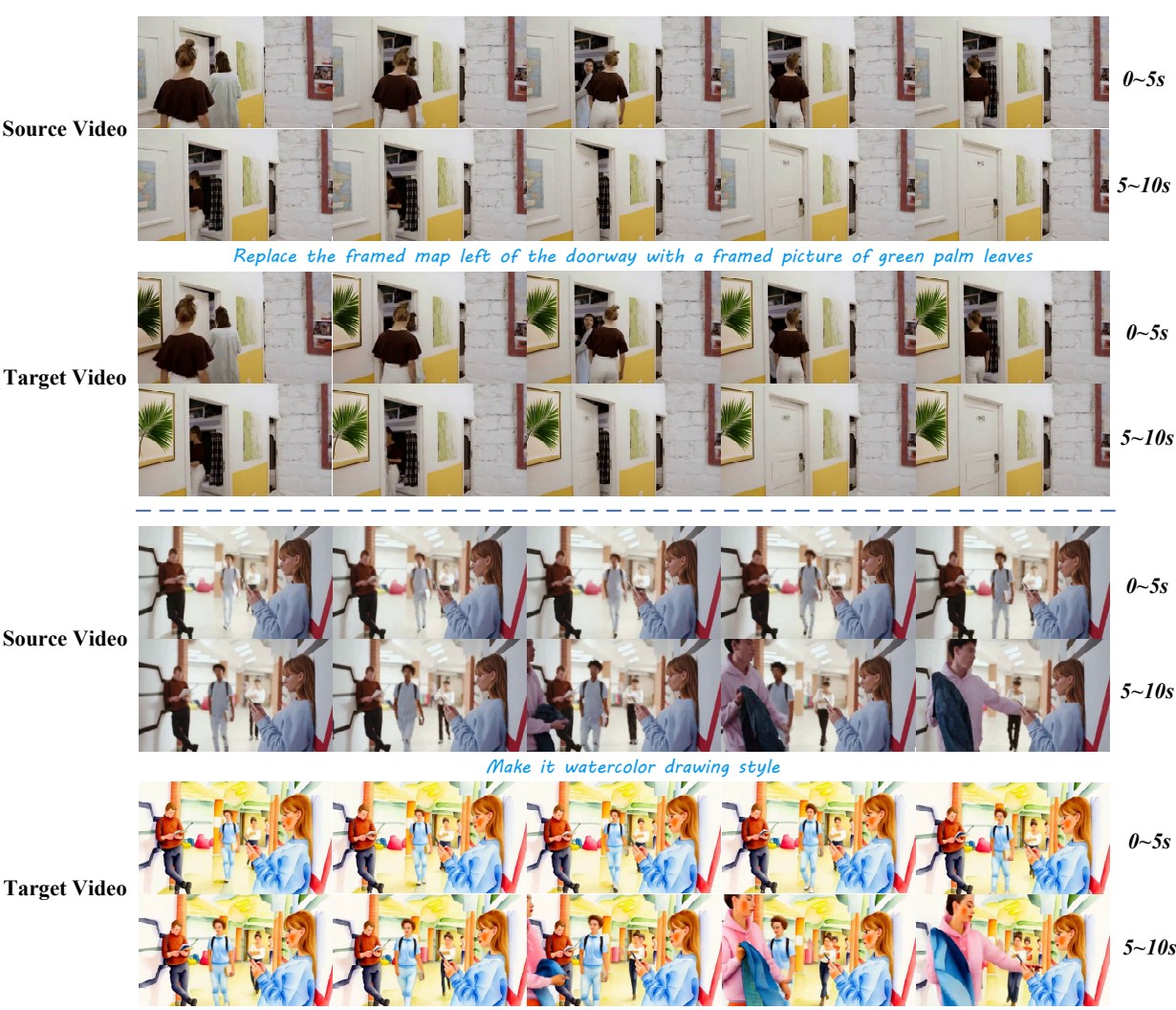

*Figure 16.* Examples of long-video editing results for replacement and stylization tasks.

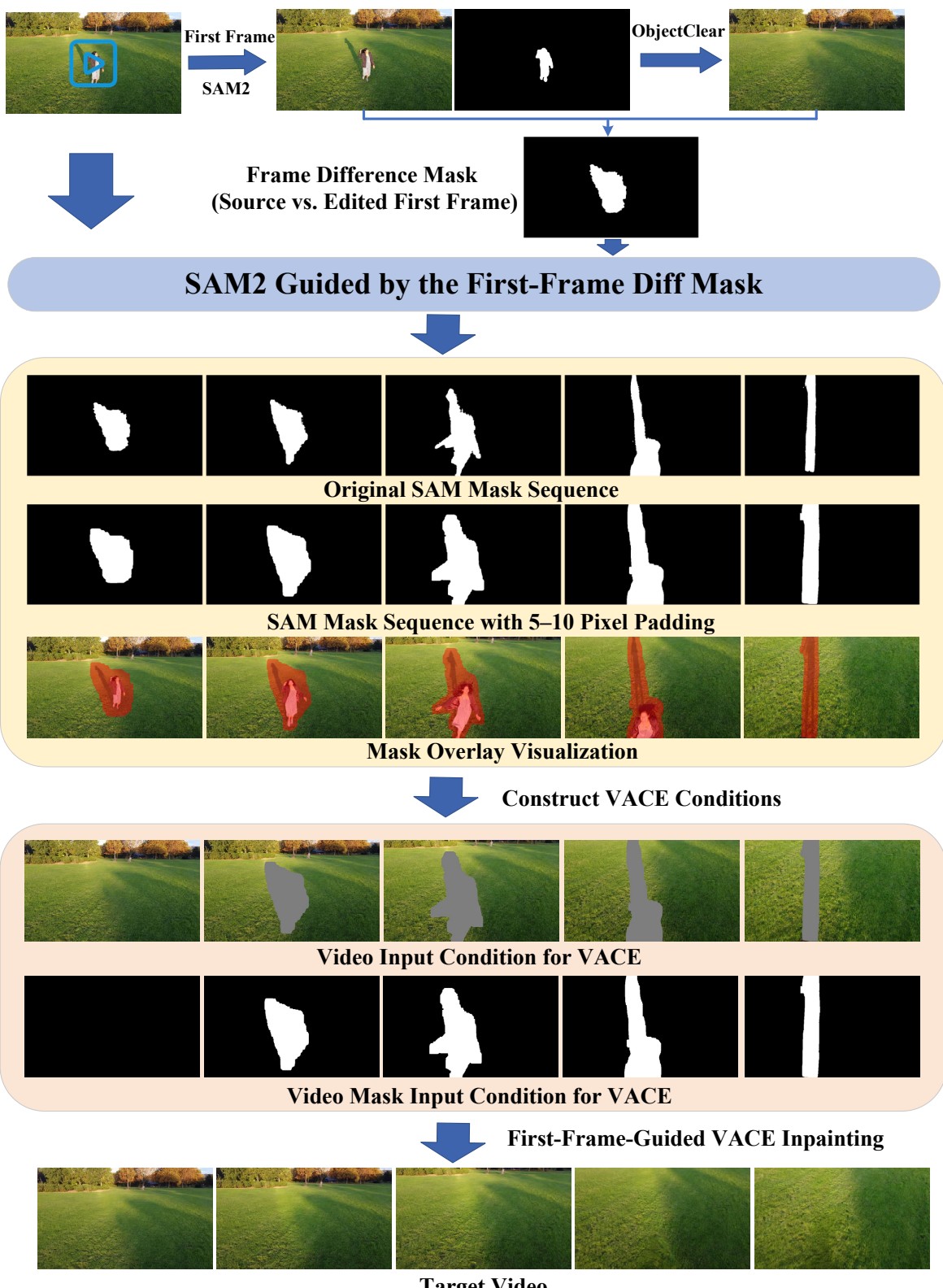

*Figure 17.* Overview of the data construction pipeline for the object removal task.

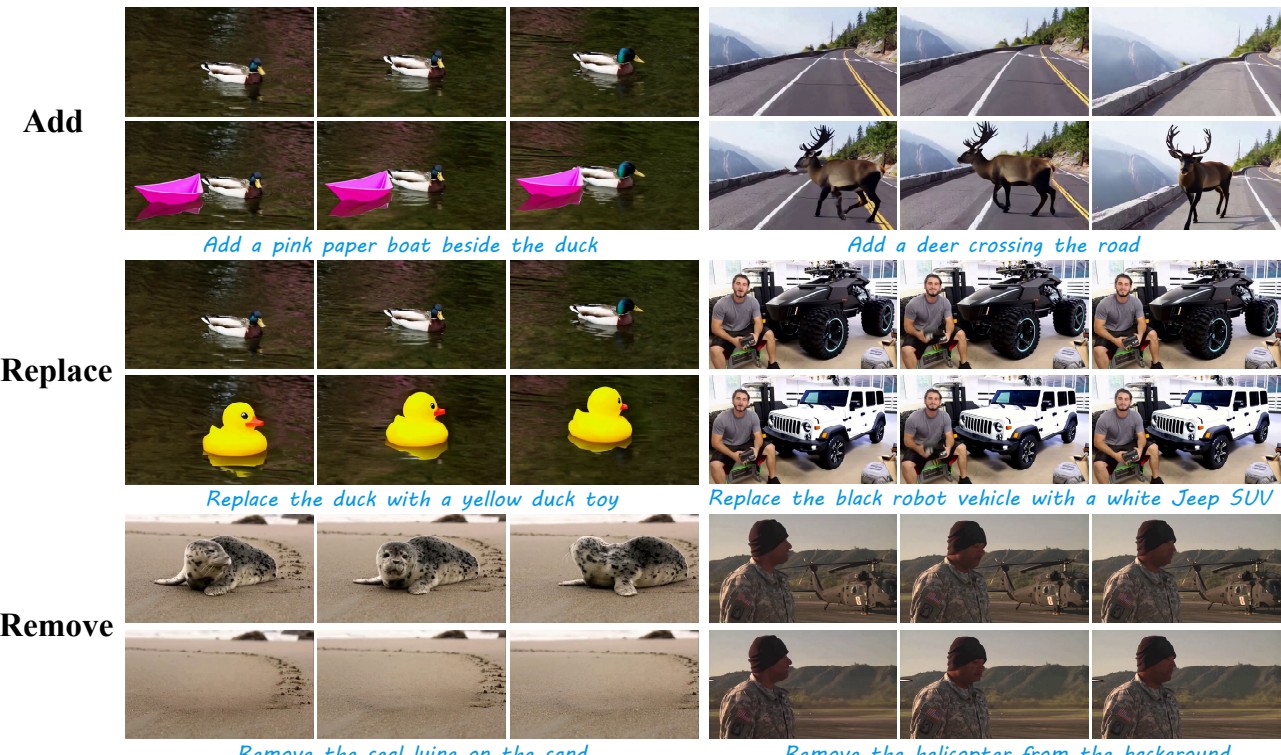

Add

Add a pink paper boat beside the duck

Add a deer crossing the road

Replace

Replace the duck with a yellow duck toy

Replace the black robot vehicle with a white Jeep SUV

Remove

Remove the seal lying on the sand

Remove the helicopter from the background

*Figure 18.* Examples of edit results across three tasks, including object deletion, replacement, and addition, highlighting the resulting shadow and reflection effects. For each example, the top row shows the original video and the bottom row shows the edited video.

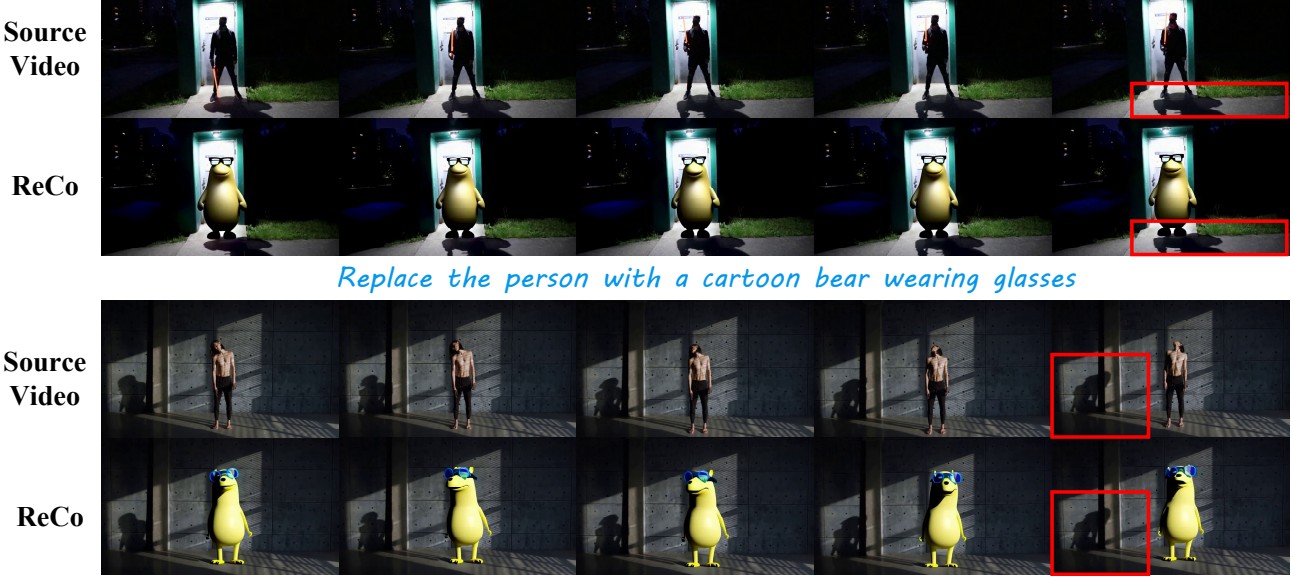

Source Video

ReCo

Replace the person with a cartoon bear wearing glasses

Source Video

ReCo

*Figure 19.* Failure cases on long-range shadow effects. Although our pipeline considers shadows and reflections during data construction, the model may still struggle in complex replacement scenarios, especially when the shadow is spatially separated from the object or extends over a long distance, as highlighted by the red boxes.

