# OpenReview forum: "In-Context Generation with Regional Constraints for Instructional Video Editing"
_ICML.cc/2026/Conference — ICML 2026 regular_

### Official Review · Reviewer_oagf · 2026-03-07

**Soundness:** 3
**Presentation:** 3
**Significance:** 3
**Originality:** 3
**Overall Recommendation:** 4
**Confidence:** 3

**Summary:**

This paper introduces ReCo, a novel instruction-based video editing framework that tackles the challenges of inaccurate editing localization and cross-region token interference. The authors innovatively adapt the "in-context generation" paradigm to the video domain by width-wise concatenating source and target videos for joint denoising. To achieve precise regional control during training, ReCo implements two core regularization constraints: (1) Latent-space Regularization, which maximizes the latent discrepancy in the editing region while minimizing it in non-edited areas to enforce background consistency, and (2) Attention-space Regularization, which suppresses the target editing region's attention to the corresponding source region, thereby mitigating interference from original object features and improving integration with the target background.

Additionally, to address the critical scarcity of high-quality paired training data, the authors construct and release ReCo-Data, a massive dataset comprising 500K high-quality instruction-video pairs spanning addition, removal, replacement, and stylization tasks. Extensive evaluations demonstrate that ReCo significantly outperforms existing state-of-the-art methods across both VLM-based and traditional metrics.

**Compliance With Llm Reviewing Policy:**

Affirmed.

**Final Justification:**

The authors' rebuttal addresses my concerns. Considering the novelty and contributions of this work, I vote for accepting the paper

**Key Questions For Authors:**

Computational Cost of Concatenation: Could you provide a quantitative comparison of the computational complexity (FLOPs and memory footprint) between your width-wise concatenation strategy and a channel-wise concatenation baseline? Given the doubled sequence length, how does ReCo scale when extending to higher resolutions (e.g., 720p/1080p)?

Implicit Grounding During Inference: Since the training regularization losses in latent space and attention space strictly require explicit masks $M$, exactly how does the model maintain precise region control during inference without a mask input? Providing visualizations of the cross-attention or self-attention activation maps during inference would greatly help prove that the model autonomously localizes the regions specified by the text instruction.

**Limitations:**

The authors have provided a brief "Impact Statement" addressing the potential misuse of video generation models and mitigation via content detection. However, they have not adequately discussed the technical limitations of their framework.

**Strengths And Weaknesses:**

Strengths:

Innovative In-Context Video Generation: Translating in-context learning from images to video is highly challenging due to complex spatiotemporal dynamics that often cause source and target features to entangle. The proposed dual regional constraints (latent and attention spaces) brilliantly address this. By applying regularization to the one-step backward denoised latents and carefully partitioning the attention matrix, the model is mathematically guided to "forget" the replaced source features and "remember" the preserved background.

Highly Valuable Dataset Contribution: The release of the 500K-scale ReCo-Data is a major boon to the video generation community. The data construction pipeline—incorporating aesthetic filtering, SAM2 segmentation, VACE synthesis, and VLLM recaptioning—is exceptionally rigorous and yields a much higher proportion of high-quality samples (91.6%) than prior datasets.

Robust Multi-task Unification: ReCo elegantly handles localized edits (adding, replacing, removing) and global edits (stylization) within a single, unified model architecture. Furthermore, it demonstrates impressive zero-shot generalization to abstract instructions (e.g., adding a glowing orb or smoke), proving the model has learned deep semantic alignments rather than merely memorizing data distributions.


Weaknesses:

Computational Bottlenecks of Width-wise Concatenation: ReCo operates by width-wise concatenating the source and target videos ($x_1^{ic} = [x_1^{src}, x_1^{tar}]$). In a Diffusion Transformer (DiT) architecture, doubling the spatial sequence length results in a quadratic increase in self-attention computational cost. This architectural choice may severely limit the framework's scalability to high-resolution (e.g., 1080p) or longer sequence videos. The paper lacks a discussion on this computational penalty compared to alternatives like channel-wise concatenation.

Ambiguity Regarding Mask-Free Inference: The paper claims the model operates effectively "without specifying editing regions". However, its core theoretical contributions—the latent and attention constraints (Eq. 13-15)—rely heavily on precise ground-truth object masks ($M$) during training. The manuscript lacks an in-depth analysis or visualization of how the model manages to achieve this perfect internal attention alignment and precise spatial localization during inference based solely on text prompts.

Risk of Inheriting Synthetic Data Biases: The target videos in ReCo-Data are primarily synthesized using another editing model, VACE. While VLLM filtering is applied, ReCo is essentially distilling VACE's generation priors. If VACE struggles with physically consistent text-to-video dynamics—such as complex human-object interactions, non-rigid deformations, or severe occlusions—ReCo will likely inherit these exact failure modes. The evaluation lacks analysis on physically demanding, out-of-distribution dynamic edits.

---

> ### Author Rebuttal · Authors · 2026-03-31
>
> **Q1&W1: [Computational Cost of Width-wise Concatenation]**
>
> **A:** We test the inference cost of editing videos with resolution 480×832×81 on a single A100 GPU. The table below compares the latency and peak memory among ReCo, several baselines, and Wan+ControlNet discussed in the main paper.
>
> Lucy-Edit and InsViE are faster, but their structures are not well suited for unifying diverse editing tasks, leading to poor performances. Similarly, Ditto and Wan+ControlNet, directly utilizing reference video as ControlNet condition, also struggle to support multi-task editing within one model.
>
> In contrast, ReCo reformulates video editing as an in-context generation paradigm, providing a more unified framework for multi-task video editing and yielding better performance. ReCo has only 2.1B parameters, making it one of the smallest models in the comparison and about 8.2× smaller than Ditto (17.3B). It requires only 22.8 GB of inference memory, compared with 64.3 GB for Ditto, making it much more friendly to consumer GPUs. Although ReCo has higher inference latency, it achieves the best overall trade-off between model size, memory usage, and performance. Moreover, the inference stage of ReCo can be further accelerated via model distillation, and the sampling step can be reduced to 8 or 4, potentially bringing inference time to around 1 minute.
>
>
> |Model|Add|Replace|Remove|Style|Params|Latency (s)|Peak Mem (GB)|FLOPs|
> |---|---:|---:|---:|---:|---:|---:|---:|---:|
> |InsViE | 3.05 | 3.17 | 3.16 | 7.91 | 6.9B | 126 | 35.4 | 18.3P |
> |Lucy-Edit | 6.31 | 6.72 | - | 4.83 | 5B | **86** | 29.2 | 10.6P |
> |Ditto | 7.56 | 6.58 | - | 9.01 | 17.3B | 1,023 | 64.3 | 95.9P |
> |Wan+Concate | 6.63 | 7.86 | 6.13 | 8.79 | **1.3B** | 178 | **20.3** |**8.7P** |
> |ReCo | **8.23** | **8.74** | **7.00** | **9.17** | 2.1B | 533 | 26.9 | 90.6P |
>
>
>
> **Q1: [Scaling to 720p/1080p Video Editing]**
>
> **A:** Scaling video generation to 720P or 1080P is an inherent challenge for nearly all video generation models. For example, Wan-2.1 only supports 480P and 720P, and its 1.3B version supports only 480P. If the model has not seen higher-resolution data during training, directly scaling video generation to 720P usually lead to unsatisfactory performance, as also shown in [Figure D](https://reco-supp.github.io/reco_page/#figure-d). Thus, when directly extending video editing to 720P, we also observe clear quality degradation for ReCo. It is also worth noting that, given the high cost of high-resolutions video generation, directly collecting and training on 720p data involves significant overhead. Therefore, a more practical solution is using a cascaded pipeline: first generating or editing the video at a lower resolution, and then applying super-resolution to obtain the final high-resolution output.
>
>
>
> **Q2&W2: [Implicit Grounding during Inference]**
>
> **A:** The mask supervision used during training enables ReCo to learn more accurate region localization. Specifically, the regional constraint loss encourages the model to allocate greater attention to the regions that should be edited. As a result, during inference, ReCo can directly identify the target editing regions and their associated attention regions without requiring an explicit mask input. We view this as evidence that ReCo has developed an implicit grounding capability. To verify this behavior, we provide visualization results in [Figure E](https://reco-supp.github.io/reco_page/#figure-e).
>
>
>
> **W3: [Risk of Inheriting Synthetic Data Biases]**
>
> **A:** We agree that ReCo may exhibit certain physical inconsistencies, which can be attributed to two primary factors. First, such inconsistencies are a prevalent challenge in current video generation research, rather than a flaw unique to our approach. Second, ReCo may inherit specific physical biases from the VACE-based data generation pipeline. However, we believe such impact can be mitigated by our design, as our model is constructed on VACE, ensuring better compatibility between training data and model optimization.
>
> As suggested, we use the Physics Adherence from WorldModelBench [2] as a evaluator to measure the physical consistency in video editing. We apply it to all baselines and our model across all tasks. The results show that ReCo achieves better physical consistency than existing baselines. We believe the physics adherence be further improved by using stronger video generation backbones, incorporating real video data into training (e.g., joint training on T2V and V2V data), and constructing higher-quality data with better physical interactions.
>
>
> [2] Li, Dacheng, et al. "WorldModelBench: Judging Video Generation Models as World Models." arXiv preprint arXiv:2502.20694 (2025).
>
> | Metric | ReCo | Ditto | InsViE | Lucy-Edit |
> | :--- | :---: | :---: | :---: | :---: |
> | Physics Adherence | **4.61** | 4.58 | 4.38 | 4.33 |

---

> > ### Author Rebuttal · Reviewer_oagf · 2026-04-02
> >
> > The authors' reponse address my concerns. I keep my score.

---

> > > ### Author Response · Authors · 2026-04-08
> > >
> > > Thank you very much for reading our rebuttal. We are glad to hear that your concern has been addressed. We believe that this discussion will further strengthen the paper. Thank you again for your thoughtful comments.

---

### Official Review · Reviewer_td1h · 2026-03-12

**Soundness:** 3
**Presentation:** 3
**Significance:** 3
**Originality:** 3
**Overall Recommendation:** 4
**Confidence:** 3

**Summary:**

This paper introduces ReCo, a framework for instruction-based video editing that reformulates the task as an in-context generation problem by jointly denoising width-wise concatenated source and target videos.
To address the challenges of precise localization and cross-region token interference without explicit masks, the authors propose two regional constraints in the latent and attention spaces.
Specifically, latent regularization enforces background consistency and amplifies editing region modifications , while attention regularization reduces token interference from the original content to better harmonize new objects with the background. Additionally, the paper contributes ReCo-Data, a large-scale dataset containing 500K instruction-video pairs across four editing tasks (addition, removal, replacement, and stylization) to facilitate future research.

**Compliance With Llm Reviewing Policy:**

Affirmed.

**Final Justification:**

The authors' rebuttal addresses my concerns. I will keep my score.

**Key Questions For Authors:**

+ The latent and attention constraints rely on a binary mask $M$ and predefined regions (A1, A2, A3) during training. Since the model is purely instruction-based at inference, does the model implicitly learn to predict these regions internally, or does the in-context design naturally guide the attention without needing explicit regions during inference?
+ Width-wise concatenation of the source and target videos effectively doubles the spatial sequence length for the DiT backbone. Could you provide a breakdown of the training memory and inference latency impacts compared to standard conditioning methods like ControlNet?
+ Both your training data filtering and the primary evaluation benchmark utilize Gemini. How do you mitigate the risk that the VLLM evaluator simply prefers outputs that align with its own curation biases rather than objective human preference, beyond the limited 10-person human study provided?

**Limitations:**

No. The author should discuss the increased computational complexity of the in-context concatenation approach , the reliance on automated segmentation pipelines for training masks , and the potential biases introduced by heavily relying on VLLMs for both dataset curation and the evaluation benchmark.

**Strengths And Weaknesses:**

**Strengths**

- Treating instructional video editing as an in-context generation task with joint denoising is a clever adaptation of recent image-editing techniques.
- Latent and attention regional constraints address mask-free localization and token interference issues in video editing.
- ReCo-Data, with 500K high-quality pairs, directly tackles the data scarcity problem in training-based video editing.

**Weaknesses**

- High computational overhead. The in-context generation approach doubles the spatial sequence length, but trade-offs in memory and inference latency are not fully discussed.
- Dependence on pre-computed masks. The attention and latent constraints rely on a binary mask and region partitions, making training sensitive to the quality of the segmentation pipeline.
- Potential evaluation bias. Heavy reliance on Gemini for dataset filtering and benchmarking may favor outputs aligned with its curation distributions.

---

> ### Author Rebuttal · Authors · 2026-03-31
>
> **W1&Q2: [Memory and Latency Overhead of In-Context Generation]**
>
> **A:** We benchmark the inference cost of editing videos with resolution 480×832×81 on a single A100 GPU. The table below compares the latency and peak memory among ReCo, several baselines, and Wan+ControlNet discussed in the main paper.
>
> Lucy-Edit and InsViE are faster, but their architectures are not well suited for unifying diverse editing tasks, which leads to poor performances. Similarly, Ditto and Wan+ControlNet, directly exploiting reference video as ControlNet condition, also struggle to support multi-task editing within one model.
>
> In contrast, ReCo reformulates video editing as an in-context generation paradigm, providing a more unified framework for multi-task video editing and yielding better performance. ReCo has only 2.1B parameters, making it one of the smallest models in the comparison and about 8.2× smaller than Ditto (17.3B). It requires only 22.8 GB of inference memory, compared with 64.3 GB for Ditto, making it much more friendly to consumer GPUs. Although ReCo has somewhat higher inference latency, it achieves the best overall trade-off between model size, memory usage, and performance. In addition, the inference of our ReCo can be further accelerated via model distillation, and the sampling step can be reduced to 8 or 4, potentially bringing inference time to around 1 minute.
>
> Regarding training efficiency, ReCo converges in about 6 days on 24 A100 GPUs, using 66.2 GB training memory at a global batch size of 24. By comparison, Wan+ControlNet requires 48 GB of training memory.
>
>
> | Model | Add | Replace | Remove | Style | Params | Latency (s) | Peak Mem (GB) | FLOPs |
> |---|---:|---:|---:|---:|---:|---:|---:|---:|
> | InsViE | 3.05 | 3.17 | 3.16 | 7.91 | 6.9B | 126 | 35.4 | 18.3P |
> | Lucy-Edit | 6.31 | 6.72 | - | 4.83 | 5B | **86** | 29.2 | **10.6P** |
> | Ditto | 7.56 | 6.58 | - | 9.01 | 17.3B | 1,023 | 64.3 | 95.9P |
> | Wan+ControlNet|7.43|8.07|6.58|8.73|**2.1B**|280|**22.8**| 14.7P|
> | ReCo | **8.23** | **8.74** | **7.00** | **9.17** | **2.1B** | 533 | 26.9 | 90.6P |
>
>
> **Q1: [Implicit Region Grounding during Inference]**
>
> **A:** Yes. ReCo only uses masks during training and requires no explicit mask in the inference stage. Specifically, the regional constraint loss in our ReCo encourages the model to allocate greater attention to the regions that should be edited. As a result, during inference, ReCo can directly identify the target editing regions and their associated attention regions for content modification. We view this as evidence that ReCo has developed an implicit grounding capability. The results are demonstrated in [Figure E](https://reco-supp.github.io/reco_page/#figure-e).
>
>
> **W2: [Sensitivity to Mask Quality]**
>
> **A:** Thanks for this point. In the construction of training data, we utilize segmentation masks to generate Ground Truth (GT) data via VACE, which ensures that the masks are precisely aligned with the content. Besides, we also apply random pixel padding (e.g., 5 to 10 pixels) to the masks in training to prevent our model from overfitting to overly precise boundaries, further enhancing the model robustness. In our ReCo, the training mask serves more like a regional cue indicating where to edit, instead of a strict pixel-level supervision signal for inpainting. Consequently, ReCo is insensitive to the precision of the input masks.
>
>
>
> **W3&Q3: [Potential Evaluation Bias]**
>
> **A:** Thanks for pointing this out. We do agree that the data bias can influence the fairness of evaluation. Nevertheless, in our work, the data filtering and benchmark evaluation are two distinct pipelines with totally different system prompts. Specifically, our data filtering only requires a binary-classification judgment on whether the editing task is successfully completed, while ReCo-Bench is much more fine-grained, covering 3 major dimensions, 9 sub-dimensions, and 4 tasks. Even using the same VLMs in such two pipelines, the data bias is different and the influence is minimal.
>
> Considering the potential impacts, we further evaluate ReCo using four additionally widely-used VLMs: Gemini-2.5-Flash-Thinking, GPT-4o, Kimi-k2.5, and Qwen-3.5-plus. As shown in [Table 1](https://reco-supp.github.io/reco_page/#table-i), the overall trends are highly consistent across these evaluators, suggesting that the evaluation bias is negligible. The results are also aligned with the 10-person human evaluation in the supplementary material, further supporting the validity of our benchmark.

---

> > ### Author Rebuttal · Reviewer_td1h · 2026-04-02
> >
> > This has resolved my concern. I will keep the score.

---

> > > ### Author Response · Authors · 2026-04-08
> > >
> > > Thank you for your thoughtful and valuable feedback on our paper. We are glad that our response has resolved your concern. We believe that this discussion will further strengthen our paper, and we sincerely appreciate your comments.

---

### Official Review · Reviewer_Vo1s · 2026-03-12

**Soundness:** 2
**Presentation:** 2
**Significance:** 3
**Originality:** 3
**Overall Recommendation:** 3
**Confidence:** 4

**Summary:**

The paper addresses the challenging task of instruction-based video editing without the need for explicitly provided masks during inference. The paper proposes ReCo, a framework that formulates video editing as an in-context generation problem by width-wise concatenating the source and target videos for joint denoising. To address the inherent challenges of localization and token interference in this paradigm, ReCo introduces two region-based constraints during training: (1) Latent-space regularization, which forces a high latent discrepancy in the editing region and preserves the non-editing background ; and (2) Attention-space regularization, which suppresses the attention of the target editing tokens to their source counterparts to mitigate content interference, while encouraging attention to the target background for coherence.

**Compliance With Llm Reviewing Policy:**

Affirmed.

**Final Justification:**

I appreciate the authors' effort, which has resolved a portion of my concerns. However, I still hold fundamental reservations regarding the paper's theoretical justification and overall novelty. While I acknowledge the practical value of the introduced dataset, the methodological contribution appears somewhat incremental. I still keep my rating.

**Key Questions For Authors:**

1. The impact of ReCo-Data heavily depends on its public availability. The authors should explicitly state their intentions regarding open-sourcing the dataset and the data generation scripts.

2. The more detailed of the ReCo-Data should be discussed with other dataset.

3.some relevant study should be added to discuss
[1].FFP-300K: Scaling First-Frame Propagation for Generalizable Video Editing

**Limitations:**

yes

**Strengths And Weaknesses:**

Strengths
Adapting the in-context learning paradigm to video editing is an  solution to preserve deep semantic alignment between the source and target. The integration of spatial regional constraints, specifically the dual constraints in both latent and attention spaces, is well-motivated, intuitive addresses the token interference issue common in diffusion-based editing.

Weaknesses
1. The in-context generation paradigm requires width-wise concatenation of the source and target videos. For an already computationally heavy video Diffusion Transformer (DiT), doubling the spatial sequence length inherently increases the memory footprint and FLOPs significantly. The paper currently lacks an analysis of training/inference time, memory consumption, and computational overhead compared to non-concatenated baselines (like ControlNet-based conditioning).

2.The latent-space regularization heavily relies on a binary mask M indicating the editing region. While this mask is automatically generated via segmentation models (SAM 2) during dataset construction, the paper does not discuss how robust the training process is to noisy, inaccurate, or temporally inconsistent masks.

3.The primary concern with this paper is the limited technical innovation. The core methodology relies heavily on straightforward adaptations of existing techniques. Formulating editing as an in-context generation task by spatially concatenating inputs has been extensively explored. Extending this spatial concatenation mechanism directly to the video domain is an incremental engineering adaptation rather than a fundamental algorithmic contribution.

---

> ### Author Rebuttal · Authors · 2026-03-31
>
> **W1: [Computational Overhead of In-Context Generation]**
>
> **A:** We report the inference cost of synthesizing 480×832×81 videos on a single A100 GPU, and compare ReCo with two non-concatenated baselines, i.e., Ditto and the variant Wan+ControlNet.  Wan+ControlNet performs worst despite being built on a relatively small 1.3B model. Ditto, which applies ControlNet to a much larger 14B backbone, yields only limited improvement while substantially increasing the total parameter count to 17.3B, along with 1,023s latency and 64.3 GB peak memory.
>
> In contrast, ReCo reformulates video editing as in-context generation, which better unifies multiple editing tasks and leads to better overall performance. With only 2.1B parameters, ReCo is among the smallest models in the comparison and is about 8.2× smaller than Ditto. It requires only 26.9 GB of memory during inference, making it considerably more practical for consumer-grade GPUs. Although ReCo is slower than the lightest baseline, it provides the best overall trade-off among performance, model size, memory efficiency, and task unification. Additionally, the inference cost could be further reduced through distillation by shortening the sampling steps to 8 or 4.
>
> Regarding training efficiency, ReCo converges in about 6 days on 24 A100 GPUs. 66.2 GB GPU memory is comsumed with the global batch size of 24. In comparison, the ControlNet-based architecture generally requires 48 GB of memory during training.
>
> |Model|Add|Replace|Remove|Style|Params|Latency(S)|Peak Mem(GB)|FLOPs|
> |-|-|-|-|-|-|-|-|-|
> |Ditto|7.56|6.58|-|9.01|17.3B|1,023|64.3|95.9P|
> |Wan+ControlNet|7.43|8.07|6.58|8.73| **2.1B** | **280**|**22.8**| **14.7P** |
> |ReCo|**8.23**|**8.74**|**7.00**|**9.17**|**2.1B**|533|26.9| 90.6P |
>
>
>
> **W2：[Robustness to Imperfect Training Masks]**
>
> **A:** Thanks for this point. In the construction of training data, we utilize segmentation masks to generate Ground Truth (GT) data via VACE, which ensures that the masks are precisely aligned with the content. Besides, we also apply random pixel padding (e.g., 5 to 10 pixels) to the masks in training to prevent our model from overfitting to overly precise boundaries, further enhancing the model robustness. In our ReCo, the training mask serves more like a regional cue indicating where to edit, instead of a strict pixel-level supervision signal for inpainting. Consequently, ReCo is insensitive to the precision of the input masks.
>
>
>
> **Q1&Q2：[ReCo-Data Availability, Details]**
>
> **A:**  We strongly value openness and reproducibility, and we commit to releasing both the dataset and the data construction scripts. As mentioned in Sec. 2, Sec. 5.1, and Figure 3 of the main paper, compared with existing datasets, our ReCo-Data has a more balanced distribution across the four editing tasks (add, remove, replace, and stylization) and a higher high-quality ratio of 91.6%. More construction details are provided in Sec. A of the supplementary material . We apply a strict data cleaning pipeline, with Gemini-2.5-Flash-Thinking used at multiple stages for quality control. We believe such high-quality dataset will be valuable to the research community.
>
>
>
> **Q3：[Relevant Study]**
>
> **A:** Thanks for the suggestion. FFP-300K is a high-quality 300K dataset for video editing, featuring 720p samples and broad task coverage. ReCo-Data, by contrast, contains 500K samples focused on four most common video editing tasks, offering more samples per task on average. Overall, the two datasets reflect different design priorities and suit different application scenarios. We will discuss it in the related work section, and include more discussion of related datasets that appeared during the review period.
>
>
>
> **W3：[Main Contributions]**
>
> **A:** Though recent advances have explored in-context generation paradigm in image editing, the success of our work is more than the simple adaptation in video domain. Our ReCo addresses the key challenge of feature entanglement when shaping in-context learning for video generation. In particular, we dig into the problem by exploring two regional constrains in model optimization: 1) latent-space regularization. To alleviate unexpected content generation in editing area, the latent discrepancy of the editing region between source and target videos is emphasized while that of non-editing areas is reduced. 2) attention-space regularization. We suppresses the attention of tokens in the editing region to tokens in the same part of source video, which alleviates the interference from original editing region tokens to novel object generation. Beyond the model design, we meticulously construct ReCo-Data, a large-scale, high-quality video editing dataset comprising 500K instruction–video pairs that span a diverse range of editing tasks. We also develop ReCo-Bench, a VLM-based benchmark for evaluating video editing performance from multiple perspectives, thereby supporting both training and evaluation within the research community.

---

> > ### Author Rebuttal · Reviewer_Vo1s · 2026-04-03
> >
> > I appreciate the authors' effort in preparing the rebuttal, which has resolved a portion of my concerns. However, I still find the methodological contribution to be somewhat incremental, though I acknowledge the practical value of the introduced dataset.
> >
> > On the experimental side, it is not entirely clear how much the mask shift operation  in training  influences the model's robustness during inference. It would be helpful to see further empirical evidence illustrating this behavior. For instance, does applying a larger mask shift lead to a degradation in the final results?

---

> > > ### Author Response · Authors · 2026-04-07
> > >
> > > **A1:** Thank you for highlighting the contribution of our dataset. We also highly value the potential impact of this work on the community. Beyond supporting our current experiments in the 480P, 81-frame setting, the released ReCo-Data construction pipeline is designed to be scalable. We plan to extend it to higher-resolution and longer videos, enabling future versions of ReCo-Data with 720P resolution and longer durations.
> > >
> > >
> > > **A2:** Thank you for the helpful comment. In general, the edited-region mask used in our regional constraint during training covers the entire edited region, i.e., the union of the source object and the target object. To mitigate noisy boundaries introduced by segmentation models (e.g., SAM2), we apply random padding of 5–10 pixels by default, following the common practice in existing inpainting settings.
> > > To further study the sensitivity to mask size and mask accuracy, we evaluate several mask variants, as illustrated in [Fig. F](https://reco-supp.github.io/reco_page/#figure-f). The results in the table below show that performance remains very similar when the padding is 5–10 pixels or 10–15 pixels. It supports our hypothesis that ReCo training does not require highly accurate masks. Instead, the regional constraint mainly helps the model learn an implicit localization of the editing region. However, when the padding becomes too large (e.g., 55–60 pixels), more background objects that should otherwise be preserved are incorrectly included in the edit area. This issue is particularly harmful for small-object editing, as shown in [Fig. G](https://reco-supp.github.io/reco_page/#figure-g). When trained with such regional constraints, the model tends to modify content outside the intended editing region, and inaccurate localization may undesirably affect surrounding areas.
> > >
> > >
> > > | Model          | Add  | Replace | Remove | Style |
> > > |----------------|------|---------|--------|-------|
> > > | Reco (5–10)    | **8.23** | **8.74** | 7.00   | **9.17** |
> > > | Reco (10–15)   | 8.19 | 8.63    | **7.04** | 9.13  |
> > > | Reco (55–60)   | 7.96 | 8.12    | 6.63   | 9.08  |

---

### Official Review · Reviewer_DkWP · 2026-03-13

**Soundness:** 3
**Presentation:** 3
**Significance:** 3
**Originality:** 2
**Overall Recommendation:** 4
**Confidence:** 4

**Summary:**

This submission aims to focus on improving instructional video editing by formulating it as an in-context generation problem with explicit regional constraints therfore they propose ReCo, a framework that concatenates the source and target videos for joint denoising within a DiT.

The core technical contribution is the introduction of two regional constraints during training: (1) a latent-space regularization that maximizes discrepancy in the editing region while minimizing it in the non-editing background, and (2) an attention-space regularization that suppresses attention from the target's editing region to the source's editing region, encouraging the generation of novel content that is coherent with the target's own background.

Additionally, the authors introduce ReCo-Data, a large-scale high-quality dataset for instructional video editing.

**Compliance With Llm Reviewing Policy:**

Affirmed.

**Final Justification:**

My concerns are partially solved, so I would like to keep my score.

**Key Questions For Authors:**

1. Could you provide a quantitative comparison of the inference latency and peak GPU memory consumption between ReCo and the baseline models for an 81-frame video?
2. How sensitive is the model to imperfection the training masks?
3. Can it be extended to longer videos? High motion intensity videos?

**Limitations:**

yes.

**Strengths And Weaknesses:**

Strengths:

1. The formulation of instructional video editing as a region-constrained in-context generation task is interesting and intuitive.
2. Providing a large-scale high-quality dataset, good for the community.

Weakness:
1. The in-context generation paradigm, by concatenating the source and target videos width-wise, effectively doubles the spatial sequence length fed into the DiT, which is not fully discussed.
2. The regional constraints rely entirely on the binary masks generated during the ReCo-Data construction phase, segmentation boundaries might be flickering or not precise, which hampers performance.
3. Relightening is not considered. When a new object is added or replaced, it should naturally cast shadows or cause reflections on the surrounding background, this scenario is not discussed in the technique.
4. Qualitative samples shown in the paper are simple. How about more complex scenarios?

---

> ### Author Rebuttal · Authors · 2026-03-31
>
> **Q1&W1: [Computational Efficiency of In-Context Generation]**
>
> **A:** As suggested, we evaluated the inference cost of 480×832×81 video generation on a single A100 GPU. The table compares ReCo with baselines in terms of latency and peak memory usage. As shown in the table, Lucy-Edit and InsViE are faster, but their architectures are struggle to unify multi-task editing, whose performances are inferior. Ditto, directly using ControlNet to condition on the input video, also faces the difficulty to unify multiple editing tasks within one model. The model size (17.3B parameters) is much larger with higher latency (1,023 s), and larger peak memory (64.3 GB). In contrast, ReCo reformulates video editing as in-context generation, better supporting unified multi-task editing and leading to better overall performances. With only 2.1B parameters, ReCo is the smallest models (about 8.2× smaller than Ditto) in the table. It requires only 26.9 GB peak memory in inference, which is much more friendly to lower-resource GPUs.
>
> Although ReCo is slower than some baselines, it provides the best overall trade-off among performance, model size, memory usage, and task unification. Moreover, the inference time could be potentially reduced via model distillation, e.g., by shortening sampling process to 8 or 4 steps.
>
>
> |Model|Add|Replace|Remove|Style|Params|Latency(S)|Peak Mem(GB)|
> |-|-:|-:|-:|-:|-:|-:|-:|
> |InsViE|3.05 |3.17 |3.16 |7.91 |6.9B |126 |35.4 |
> |Lucy-Edit|6.31|6.72| - |4.83 | 5B | **86** |29.2|
> |Ditto|7.56|6.58 | - | 9.01 |17.3B |1,023 |64.3 |
> |ReCo|**8.23**|**8.74**|**7.00**|**9.17**| **2.1B** |533|**26.9**|
>
>
>
> **Q2&W2: [Sensitivity to Imperfect Training Masks]**
>
> **A:** Thanks for this point. In the construction of training data, we utilize segmentation masks to generate Ground Truth (GT) data via VACE, which ensures that the masks are precisely aligned with the content. Besides, we also apply random pixel padding (e.g., 5 to 10 pixels) to the masks in training to prevent our model from overfitting to overly precise boundaries, further enhancing the training robustness. In our ReCo, the training mask serves more like a regional cue indicating where to edit, instead of a strict pixel-level supervision signal for inpainting. Consequently, ReCo is insensitive to the precision of the input masks.
>
>
>
> **W3: [Shadows and Reflections in Video Editing]**
>
> **A:** Thanks for the insightful point. We have indeed considered relighting effects, such as shadows and reflections, during dataset construction. For the tasks of object adding or removal, we first edit the first frame using ObjectClear [1] to remove the target object together with its associated lighting effects. We then identify the changed regions, track them across frames with SAM2, and apply relatively large mask padding to better cover shadows and reflections. By using these masks and the edited first-frame, VACE can propagate the shadow/reflection changes into the subsequent frames. For object replacement, we similarly found that using a larger padded region works well in practice to synthesize relighting effects, so we adopt this simple strategy in our current pipeline.
>
> As a potential future direction, the pipeline could be improved by using stronger image editing models that better maintain illumination consistency in first-frame editing. Training on such data enables our model to learn plausible shadow- and reflection-related changes, which are presented in [Figure A](https://reco-supp.github.io/reco_page/#figure-a).
>
>
> [1] Zhao, Jixin, et al. “ObjectClear: Complete Object Removal via Object-Effect Attention.” arXiv preprint arXiv:2505.22636 (2025).
>
>
>
> **Q3&W4：[Complex Editing Scenarios]**
>
> **A:** Thanks. We provide two more challenging scenarios, i.e., video editing with large motion and multiple objects in [Figure B](https://reco-supp.github.io/reco_page/#figure-b). Even in such challenging settings, our ReCo can still perform stable video editing.
>
>
>
> **Q3：[Extension to Longer Video Editing]**
>
> **A:** The key challenge in long-video editing is maintaining temporal appearance and motion coherence across clips within the edited regions. Since our ReCo is fine-tuned based on VACE, it can naturally support temporal extension for long video editing. Specifically, we split the long video into overlapping clips (e.g., 5s, 4s, and 4s). The first clip is edited normally. For each subsequent clip, we employ the last 16 edited frames from the previous clip as condition, where the in-context input is formed as [source video, (16 edited frames + 65 masked video frames)]. Repeating this process across all subsequent clips enables editing of long videos.
>
> As shown in [Figure C](https://reco-supp.github.io/reco_page/#figure-c), after a simple fine-tuning stage, our model can successfully perform replacement editing on a 10-second video while preserving good continuity across clip boundaries.

---

> > ### Author Rebuttal · Reviewer_DkWP · 2026-04-03
> >
> > Thank you for the rebuttal. The additional experiments address several of my previous concerns, I still have several concerns:
> >
> > 1. Dataset Open-source Commitment. I think the ReCo dataset is a strength of this work. However, the authors do not explicitly state a commitment or timeline to publicly release this dataset.
> >
> > 2. In W2's rebuttal. The "random pixel padding" strategy was not mentioned in the original text, not convincing enough. Furthermore, static padding does not solve the underlying problem of temporal flickering in dynamically generated masks, which inevitably introduces temporal noise into your regional training signal. And there is no experimental evidence support the insensitivity claim.
> >
> > 3. In W3's rebuttal. Thanks for the rebuttal cases.  But I think this problem is not fully considered in the whole process, In fact, your latent-space regularization $L_{latent}$, which strictly enforces background consistency outside the editing region, discourages the model from generating dynamic shadows or reflections on the background corresponding to newly edited objects.
> > Furthermore, this regional isolation leads to a lack of overall visual harmony. While your VLLM-based scores claim high "Naturalness", the qualitative results (e.g., the chimpanzee in figure 4 and the parrot in figure 6) clearly reveal a "pasted-on" effect.
> >
> > Two additional concerns:
> >
> > 4. Relying on a single VLLM (Gemini-2.5-flash) for the proposed quantitative benchmark introduces potential model-specific biases.
> >
> > 5. Conducting a human evaluation with only 10 participants is too small to reliably prove the perceptual superiority of your method over strong baselines.

---

> > > ### Author Response · Authors · 2026-04-07
> > >
> > > **A1:** Thanks. Reviewer Vo1s also raised a similar concern in the first round, and we made the same commitment there. We will definitely release both the dataset and the complete data construction scripts immediately upon acceptance, which we believe will benefit the research community.
> > >
> > > **A2:** Thank you for the helpful comment. Here we clarify the process clearly.
> > >
> > > **Mask for regional constraint.** The masks used in local-edit data construction may contain noisy boundaries or temporal flicker. For noisy boundaries, directly using these masks for VACE-based inpainting often causes boundary artifacts. To address this issue, we apply random padding of 5–10 pixels by default, following common practice in existing inpainting settings, so that the mask more completely covers the object and leads to better editing quality. We apologize for the confusion caused by this omission and will add this missing detail to Sec. A.4 in the revision. Additionally, inpainted cases affected by severe temporal flicker usually contain noticeable residual content from the original object and can therefore be largely filtered out by the VLM-based quality control pipeline. As mentioned in the paper, our ReCo-Data can achieve a high-quality ratio of 91.6% under human evaluation. Consistent with data construction, we use the same edited-area masks with 5–10 pixel padding by default during training, covering both the original object region and the target region.
> > >
> > > **Evaluation of mask robustness.** We further evaluate several mask variants to study sensitivity to mask size and accuracy (visualized in [Fig. H](https://reco-supp.github.io/reco_page/#figure-h)). As shown in [Table II](https://reco-supp.github.io/reco_page/#table-ii), the results are very similar for 5–10 and 10–15 pixel padding, suggesting that training is not highly sensitive to precise mask boundaries. Instead, the regional constraint mainly serves as an implicit cue for edit-region localization. When using extremely large padding, more background objects that should otherwise be preserved are incorrectly included in the edit area. This issue is particularly harmful for small-object editing, as shown in [Fig. G](https://reco-supp.github.io/reco_page/#figure-g). When trained with such regional constraints, the model tends to modify content outside the intended editing region, and inaccurate localization may undesirably affect surrounding areas. Besides, we also simulate temporal flickering by applying random erosion to the edited-area masks, and find that severe temporal flickering can lead to suboptimal results, further showing that noisy data may degrade performance.
> > >
> > >
> > > **A3:** Thanks for your insightful comment. We do agree that reflections, relighting, and shadow-related effects are widely recognized as particular challenges in image/video editing, and it is difficult to fully address them. We also make every effort to account for these factors in data construction.
> > >
> > > As illustrated in [Fig. J](https://reco-supp.github.io/reco_page/#figure-j), for object removal, we first edit the first frame with ObjectClear, which can simultaneously remove the object and its associated shadows. We then compute a diff mask against the original frame and use it to guide SAM2 tracking across the video. The resulting edited-region mask sequence is further used by VACE to remove the object together with its shadows and reflections. Since the mask covers both the object and its associated effects, these regions are treated as part of the edited area rather than the preserved background, and our latent-space regularization therefore does not suppress dynamic shadow or reflection generation for newly edited objects. For the add task, we simply reverse the original and edited videos during training while keeping the same edited-region mask, which similarly allows the model to learn such effects.
> > >
> > > For object replacement, we don't use the same pipeline because reliable shadow-/reflection-aware mask-based image replacement models are still lacking, and existing instruction-based models are often inaccurate enough for stable data construction. As a compromise, we use the original mask with larger padding to cover as much of the shadow/reflection region as possible.
> > >
> > > With the resulting add, remove, and replace training data, the model can handle some general cases, as shown in [Fig. A](https://reco-supp.github.io/reco_page/#figure-a). However, clear limitations remain for more complex scenarios, especially when shadows are long-range or far from the object, as shown in [Fig. K](https://reco-supp.github.io/reco_page/#figure-k).
> > >
> > > **A4&A5:** Thanks. We agree that evaluator bias is a valid concern, which was also raised by Reviewer td1h. As shown in [Table I](https://reco-supp.github.io/reco_page/#table-i), results from four VLM evaluators are highly consistent, suggesting negligible evaluator bias. Please also refer to our response (i.e., W3&Q3) to Reviewer td1h for more details.

---

### Decision · Program_Chairs · 2026-04-30

**Decision:**

Accept (regular)

**Comment:**

The paper introduces ReCo, reformulating instruction-based video editing as an in-context generation task by jointly denoising concatenated source and target videos. To enable mask-free inference and prevent content leakage, the authors propose latent-space and attention-space regularizations during training. A major highlight is the release of ReCo-Data, a 500K-pair dataset spanning diverse editing tasks. While reviewers noted the increased computational cost of doubling sequence lengths, the rebuttal successfully clarified inference latency and demonstrated the model's robust internal grounding. The consensus emphasizes that the framework's practical utility and the significant dataset contribution provide substantial value to the video generation community. Please include all the added experiments in the camera-ready version.